# Deciphering the interplay between biology and physics with a finite element method-implemented vertex organoid model: A tool for the mechanical analysis of cell behavior on a spherical organoid shell

Julien Laussu[1,2]*, Deborah Michel[2], Léa Magne[1,2], Stephane Segonds[1], Steven Marguet[1], Dimitri Hamel[2], Muriel Quaranta-Nicaise[2], Frederick Barreau[2], Emmanuel Mas[2,3], Vincent Velay[4], Florian Bugarin[1]*, Audrey Ferrand[2]*

1 Institut Clément Ader, Université Fédérale de Toulouse Midi-Pyrénées, Institut Clément Ader–CNRS UMR 5312 –UPS/INSA/Mines Albi/ISAE, Toulouse, France, 2 IRSD—Institut de Recherche en Santé Digestive, Université de Toulouse, INSERM, INRAE, ENVT, UPS, Toulouse, France, 3 Gastroenterology, Hepatology, Nutrition, Diabetology and Hereditary Metabolic Diseases Unit, Hôpital des Enfants, CHU de Toulouse, Toulouse, France, 4 Institut Clément Ader (ICA), Université de Toulouse, CNRS, IMT Mines Albi, INSA, ISAE-SUPAERO, UPS, Campus Jarlard, Albi, France

* julien.laussu@gmail.com (JL); florian.bugarin@univ-tlse3.fr (FB); audrey.ferrand@inserm.fr (AF)

**Data Availability Statement:** Code and Raw data availability: The software, data, standard parameter

## Abstract

Understanding the interplay between biology and mechanics in tissue architecture is challenging, particularly in terms of 3D tissue organization. Addressing this challenge requires a biological model enabling observations at multiple levels from cell to tissue, as well as theoretical and computational approaches enabling the generation of a synthetic model that is relevant to the biological model and allowing for investigation of the mechanical stresses experienced by the tissue. Using a monolayer human colon epithelium organoid as a biological model, freely available tools (Fiji, Cellpose, Napari, Morphonet, or Tyssue library), and the commercially available Abaqus FEM solver, we combined vertex and FEM approaches to generate a comprehensive viscoelastic finite element model of the human colon organoid and demonstrated its flexibility. We imaged human colon organoid development for 120 hours, following the evolution of the organoids from an immature to a mature morphology. According to the extracted architectural/geometric parameters of human colon organoids at various stages of tissue architecture establishment, we generated organoid active vertex models. However, this approach did not consider the mechanical aspects involved in the organoids' morphological evolution. Therefore, we applied a finite element method considering mechanical loads mimicking osmotic pressure, external solicitation, or active contraction in the vertex model by using the Abaqus FEM solver. Integration of finite element analysis (FEA) into the vertex model achieved a better fit with the biological model. Therefore, the FEM model provides a basis for depicting cell shape, tissue deformation, and cellular-level strain due to imposed stresses. In conclusion, we demonstrated that a combination of vertex and FEM approaches, combining geometrical and mechanical parameters, improves modeling of alterations in organoid morphology over time and enables better assessment of

files, and tutorials can be found in the GitHub repository: https://github.com/organofem. The "organofem-published" repository is dedicated to this paper and includes a complete and standalone package (full version with Python and C codes). It contains all the necessary codes, along with a guide for installing the required dependencies, with examples such as standard parameter files and a tutorial for testing the software. Documentation is also provided in the form of a README.md file to guide users. The raw data corresponding to this article can be found in the "test/article_rawdata" folder within the GitHub repository.

**Funding:** This collaborative work was funded by two Plan Cancer projects (French national grants): Mocassin - Biosystem 2017 (grant agreement number C18006BS, granted to AF) and Melchior - MIC 2020 (grant agreement number C20048BS, granted to FBu). The Biodige Protocol (NCT 02874365) was financially supported by Toulouse University Hospital. JL is supported by both Plan Cancer projects. DM is supported by Plan Cancer Melchior - MIC 2020. LM is supported by Inserm and the Région Occitanie. DH is supported by the Université Paul Sabatier and Région Occitanie. The funders had no role in study design, data collection and analysis, decision to publish, or preparation of the manuscript.

**Competing interests:** The authors have declared that no competing interests exist.

the mechanical cues involved in establishing the architecture of the human colon epithelium.

## Author summary

This study explores the interplay between biology and mechanics in tissue architecture, particularly the 3D organization of human colonic epithelial organoid. The experimental approach focused on imaging in culture the organoids for 120 hours to follow their morphological maturation. From these images, the architectural and geometric parameters of the biological organoids were extracted and used to create in silico organoid by the vertex method. However, this method did not take into account the mechanical forces involved in the morphological evolution of the organoids. To overcome this limitation, a finite element method was applied to the vertex model. Using the Abaqus solver, mechanical constraints, such as those undergone by biological organoids, were simulated. The integration of FEM into the vertex model improved the correspondence between the biological model and the modeling, providing a more accurate representation of tissue deformations, cellular forces and mechanical tensions undergone by the biological organoid. This method enabled creation of a digital model of the human colon tissue to aid in understanding the roles of mechanical forces in establishing human colon tissue architecture over time.

## Introduction

Epithelial tissues are continually evolving and must adapt to their environment. Alterations in their ability to generate or resist forces can lead to pathological conditions such as inflammatory diseases or cancer [1]. Understanding the interplay between biology and mechanics involved in tissue architecture is challenging, particularly in terms of 3D tissue organization. An appropriate biological model is required to enable multi-scale observations, ranging from cellular resolution to global tissue architecture; in addition, pertinent theoretical and computational approaches are necessary to generate a synthetic model that has the highest possible relevance to the biological model and enables investigation of the mechanical stresses on tissues.

To this end, an organoid recreating a monolayer of epithelial tissue is a suitable biological model. Organoids, established from adult stem cells, embryonic stem cells, or induced pluripotent stem cells, on the basis of their self-renewal and differentiation capabilities, are simplified versions of organs and tissues. Organoids, cultured in three-dimensional (3D) space in a supporting matrix, are autonomous and self-organized, and recapitulate the architecture and at least one function of the tissue of origin [2–9]]. These features make this model an ideal tool for studying the interplay between cell biology and physics in establishing tissue architecture. Furthermore, in vitro culture allows for live microscopic observation, which offers relatively high temporal resolution in monitoring the development of tissue architecture.

Experiments to understand monolayer epithelium homeostasis have largely focused on cell proliferation, cell differentiation, and cell death regulation [10,11]. Many studies on 2D cultures have characterized the properties of cell geometry and the fluidic transitions occurring during tissue remodeling [12,13]. The deformation of tissues can be assessed in 2D or 2.5D under experimental conditions, thus highlighting the individual contributions of cell density, cellular organization, and the composition of the extracellular matrix [14,15]. However, other tissue architecture parameters, such as cell shape regulation [16,17] or mechanical stress,

through osmotic pressure [18], external solicitation [19,20], and active contraction [21], are involved in shaping epithelial tissue A powerful computational model combining biology, physics, and geometry is necessary to decipher the individual contributions of these inter-linked cell events to building the intestinal tissue architecture.

Theoretical and computational approaches inspired by physics and combined with experimental observations of this biological model have been developed to understand the relevant physical constraints [22]. Recent synthetic models have focused on 3D processes occurring in morphogenetic events [23–25]. Among them, 3D vertex models (VMs) are well adapted to biological systems, because they take macroscopic mechanical characteristics into account while providing cell identity representation [26,27].

VMs can be considered a geometrical description of the biological system, in which each cell is represented by polygons, expressed as vertices connected with edges; discretization of this geometry is accomplished by using basic elements, for which mechanical equations are solved. The ability to represent each cell individually enables study of tissue organization. The 3D VM can be considered a base model for applying a finite element method (FEM) to examine mechanical properties [28,29]. FEM uses volumetric meshes to calculate internal stresses and forces throughout the volume of an object. This well-established numerical analysis method is used in mechanics to model the deformation of complex structures for which an analytical solution is difficult to obtain. Moreover, FEM is well suited to modeling the effects of deformation due to local constraints [30,31]. Among the commonly used software solutions in the industry, Abaqus is an effective option providing many predefined elements for studying complex physical phenomena. This software provides powerful flexibility and modularity to approximate complete partition by polynomial functions on each element. Consequently, FEM, which provides a discrete solution to a continuous problem, has frequently been used for solid tissue modeling [31,32].

Because morphology is a principal feature defining intestinal monolayer epithelial organoids [32,33], we used the volumetric VM of organoids combined with FEM to simulate modifications to the human colon organoid architecture, according to simple physical laws and material properties. Briefly, using an approach combining real-time fluorescence imaging and 2D and 3D image analysis, we extracted biological parameters characterizing the organoid biological model during the establishment of its architecture. Applying these parameters in an "organoid" VM enabled us to observe the limits of the vertex method by comparing the data obtained from the biological organoid and the in silico active VM (AVM). Finally, to obtain a more relevant in silico organoid model, and to understand the effects of the physics involved in the dynamic morphological changes contributing to establishment of the colorectal organoid architecture, we generated an innovative FEM model. A substantial part of this study was aimed at integrating the interface between the AVM and the FEM model by developing Python algorithms to translate vertex architecture into elements preexisting in Abaqus. The second part of our algorithmic development was aimed at extracting morphological information from the FEM model by recreating a surface mesh enabling measurement of cellular surface areas. Analysis of the results demonstrated how our FEM model's flexibility enables linkage of cell shape, tissue deformation, and stresses at cellular level under imposed constraints.

In conclusion, using a human colon monolayer epithelium organoid as a biological model, we generated an innovative in silico model of the human colon organoid. This model provides insights into the interplay between biology and mechanics. Moreover, the model demonstrates that combining vertex and FEM approaches can enhance modeling of the organoid morphology alterations over time and enable better assessment of the mechanical pointers involved in establishing the architecture of the human colon epithelium.

## Results

### Extraction of biological parameters from human colon organoids

Our first objective was to extract architectural/geometric parameters of human colon organoids at various stages of tissue architecture establishment. As previously noted [33,34], human colon organoids are established from crypts extracted from patients' colon biopsies. The crypts are cultivated in a Matrigel matrix to obtain colon organoids (Fig 1A). During culture, we followed the establishment of the organoid architecture and acquired stack images, which were used to reconstruct the organoid's 3D morphology and determine its parameters at cellular resolution. Table 1 provides the geometrical parameters characterizing the biological organoid model, such as their diameter; the estimated number of cells forming the organoid, according to live microscopy observation; and the organoid shell thickness, reflecting maturation stage as well as the number of cells for which individual geometrical properties were determined. The mean size of the imaged organoids was approximately 350 μm in diameter, and the mean number of cells per organoid was equivalent to 390 (Table 1). As highlighted by staining of nuclei (DRAQ5 or SpyDNA), cell membrane proteins (EpCAM), or the actin-myosin cytoskeleton (visualized by staining actin with phalloidin or FastActin), colon organoids displayed different morphologies during culture. The organoid evolved from an "immature" spherical morphology (Fig 1B, T0)—characterized by a flat epithelium with flattened nuclei and elongated cells, along with a central lumen (L)—to a "mature" morphology (Fig 1B, T0 + 180 minutes). In the mature state, the epithelium became thicker and polarized, and the area at both the apical (facing the lumen) and basal (facing the matrix) poles decreased. This maturation was also marked by elongation of the lateral sides, swelling of the apical membrane, nuclear positioning near the basal pole, a smaller central lumen, and overall deformations resulting in a less spherical organoid structure (Fig 1B, 1C and 1D). An intermediate stage (Fig 1B, T0 + 90 minutes) is referred to herein as "intermediate" morphology (Fig 1B and 1D).

Because cells are the morphological units of tissues, we considered that global morphological changes in organoids might be associated with architectural alterations at the cellular level. We thus sought to first extract information for individual cells. On the basis of the organization of the actin-myosin cytoskeleton and its fluorescence staining (Fig 1Ea), we used Cellpose 2.0 [35,36] to perform 2D segmentation (Fig 1Eb) and then performed 3D reconstruction of the organoid (Fig 1Ec). Contrary to what the image acquisition cross-section might suggest, all cells were attached to both the basal and the apical planes. Application of this approach to immature, intermediate, and mature morphological stages allowed us to extract the volume, apical and basal surface area, and lateral surface area (cell-cell surface area) for each cell, and to assess the cell aspect ratio (Fig 1F). Moreover, to incorporate topological information, we calculated a spreading distance corresponding to the mean distance between a cell centroid and the centroids of the first order neighboring cells. Because having three or four direct neighbors is a very rare event [31], we decided to measure the mean distance to the five nearest neighbors.

On the basis of these data, we developed an in silico model of the intestinal organoid.

### Generation of a vertex model for the human colon organoid

In our organoid VM, each cell is a polygonal mesh comprising a set of vertices, faces, and edges (Fig 2A). Vertices are the set of edge nodes. Faces are the external structures (i.e., the membranes) of cells for the cell-cell side (Fig 2A, light yellow), cell-basal side (Fig 2A, light right), or apical side (Fig 2A, light blue) interfaces. Edges are segments located on the perimeters of polygonal faces. Thus, to individualize each cell within the overall structure, the vertices, faces, and edges define and depend on the cell's identity.

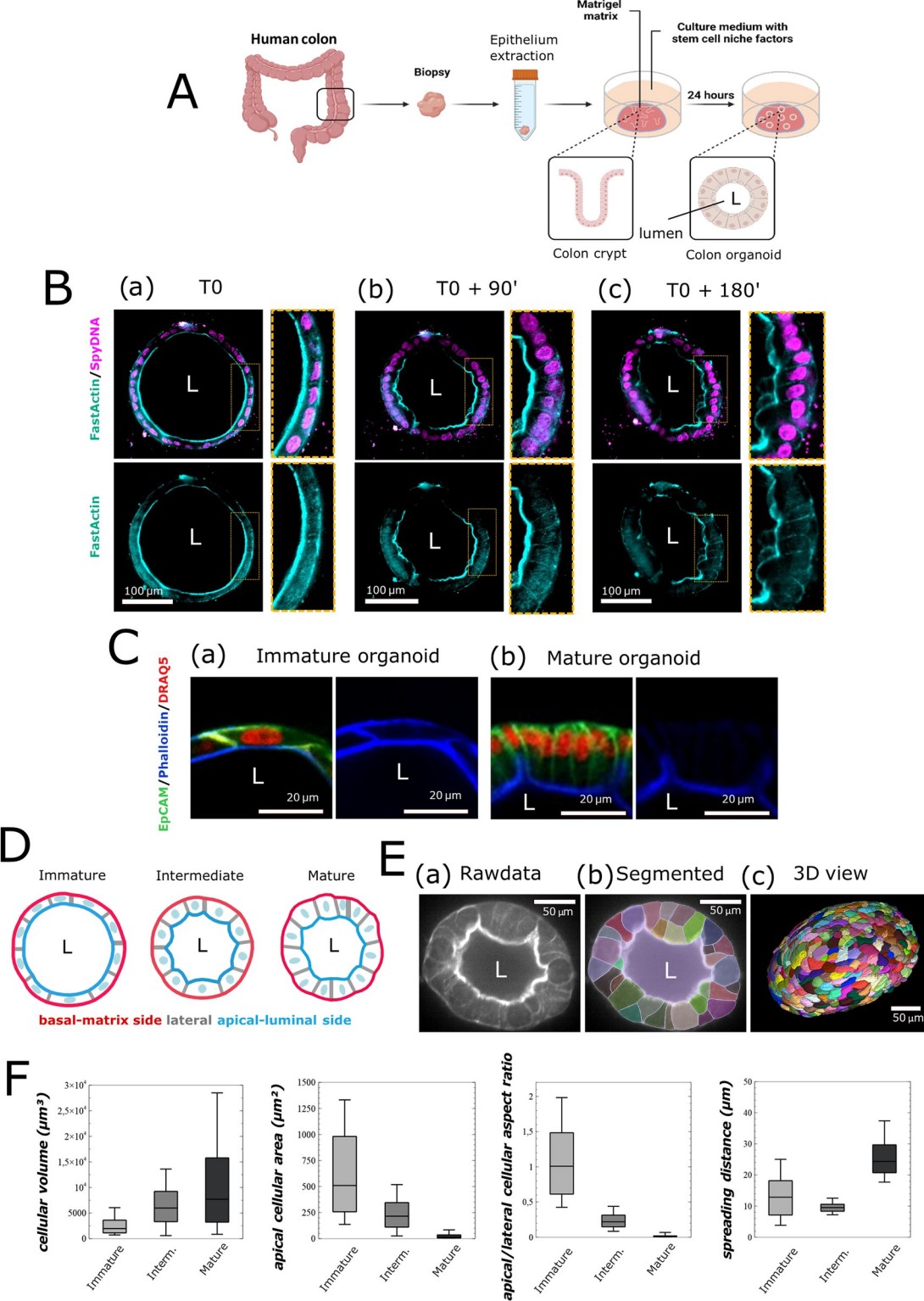

**Fig 1. Description of the biological organoid model.** A: Schema representing the experimental process used to obtain colon organoids from human biopsies after isolation of the crypts and culture in Matrigel matrix. Illustration was created with BioRender.com. B: Growth pattern of human intestinal organoids in Matrigel on a 3-hour timelapse analysis using FastActin dye (cyan) and nuclei SpyDNA dye (purple). A first acquisition of organoid presenting immature morphology (spherical structure characterized by a flat epithelium with flattened nuclei and elongated cells, along with a central lumen) is done (T0) (a). After

deflation and loss of organoid sphericity (T0 + 90') (b) in a short time window, the apical membrane becomes increasingly irregular, and cell nuclear orientation changes from parallel to perpendicular to the apical border (T0 + 180') (c). C: High magnification at cellular scale of a confocal snapshot imaged on a fixed colon organoid with phalloidin (blue) immunostaining, EpCAM (green) immunostaining, and nuclear Draq5 (red) staining, illustrating (a) an immature organoid or (b) a mature organoid after 15 days of culture in Matrigel. D: Schema illustrating the global and cellular deformation characterizing the Immature, Intermediate, and mature morphology of the previously observed living organoids. E: Presentation of the protocol used to quantify organoid morphology: (a) based on FastActin imaged on confocal microscopy, and (b) with cellular compartments segmented with a machine-learning based approach (Cellpose), in 2D and then stitched and interpolated in 3D. The lumen is manually segmented with Napari [70,77] to define the lumen/apical interface. The 3D exploration and manual correction of another larger segmented organoid were performed with Morphonet [79] or Napari [77] (c). F: Organoid morphology descriptors presented in four different distribution box plots for cellular volume, apical surface area, apical/lateral aspect ratio, and spreading distance distribution for one immature organoid, one intermediate organoid, and one mature organoid. Box and whisker plots indicate quartiles.

To construct the in silico organoid, because the initial structure of human colon organoids is spherical (Fig 1), we deposited seeds on a standardized sphere surface (Fig 2B). According to the mean number of cells in the biological organoids analyzed (Table 1), we used 400 seeds for our vertex organoid models. Next, a three-stage mesh was used to obtain cell volumes. To deposit the seeds, we used a spherical function and the "golden angle" to create a Fibonacci lattice on the surface of the normalized sphere (radius = 1). A normal deviation around this function was applied to achieve a non-uniform distribution of points. A Delaunay tessellation connected neighboring seeds, and a Voronoi tessellation delimited the cell area in the 2.5D shell. Finally, we duplicated and connected two copies of the 2.5D shell through spherical dilation, thereby forming a closed 3D monolayer tissue in which the apical and basal faces of each cell are similar in shape.

After generating the 3D volume, we applied the following dimensions to the model: the radius of the spherical structure, according to the average radius measured for each organoid stage (immature, intermediate, and mature), and the relative tissue thickness, determined as the ratio of the mean thickness normalized by the mean radius of the biological organoids at each stage (Table 1). To mimic the immature, intermediate, or mature organoid stages, we set the relative tissue thickness ratio in our in silico model to 0.07, 0.25, and 0.4, respectively (Fig 2C). These steps generated a vertex organoid model for each organoid stage (Fig 2Ca, 2Cd and 2Cg).

We then tested previously proposed AVMs on these models to study global epithelial deformation [37–39]. The earlier models used 2D solvers based on a simple energy functional equation (the mechanical behavior law governing the system) to model epithelial monolayer relaxation, as described in the Farhadifar 2D model [40].

**Table 1. Parameters extracted from live observation and analysis of the biological human colon organoid model (*"estimated" corresponds to a spherical abstraction using the perimeter to area transformation, and the thickness ratio is relative to the radius of the final organoid. Radius of the organoid/(radius basal-radius apical.).**

| Morphology | Organoid ID | Diameter (µm) | Cell number (estimated*) | Cell number (analyzed*) | Thickness ratio |
|---|---|---|---|---|---|
| Immature | 1 | 513 | 387 | 194 | 0,084 |
| Immature | 2 | 572 | 412 | 226 | 0,055 |
| Immature | 3 | 290 | 276 | 173 | 0,085 |
| Intermediate | 4 | 230 | 369 | 205 | 0,275 |
| Intermediate | 5 | 178 | 403 | 345 | 0,194 |
| Intermediate | 6 | 277 | 520 | 312 | 0,283 |
| Mature | 7 | 265 | 387 | 209 | 0,512 |
| Mature | 8 | 384 | 241 | 168 | 0,41 |
| Mature | 9 | 474 | 518 | 363 | 0,342 |

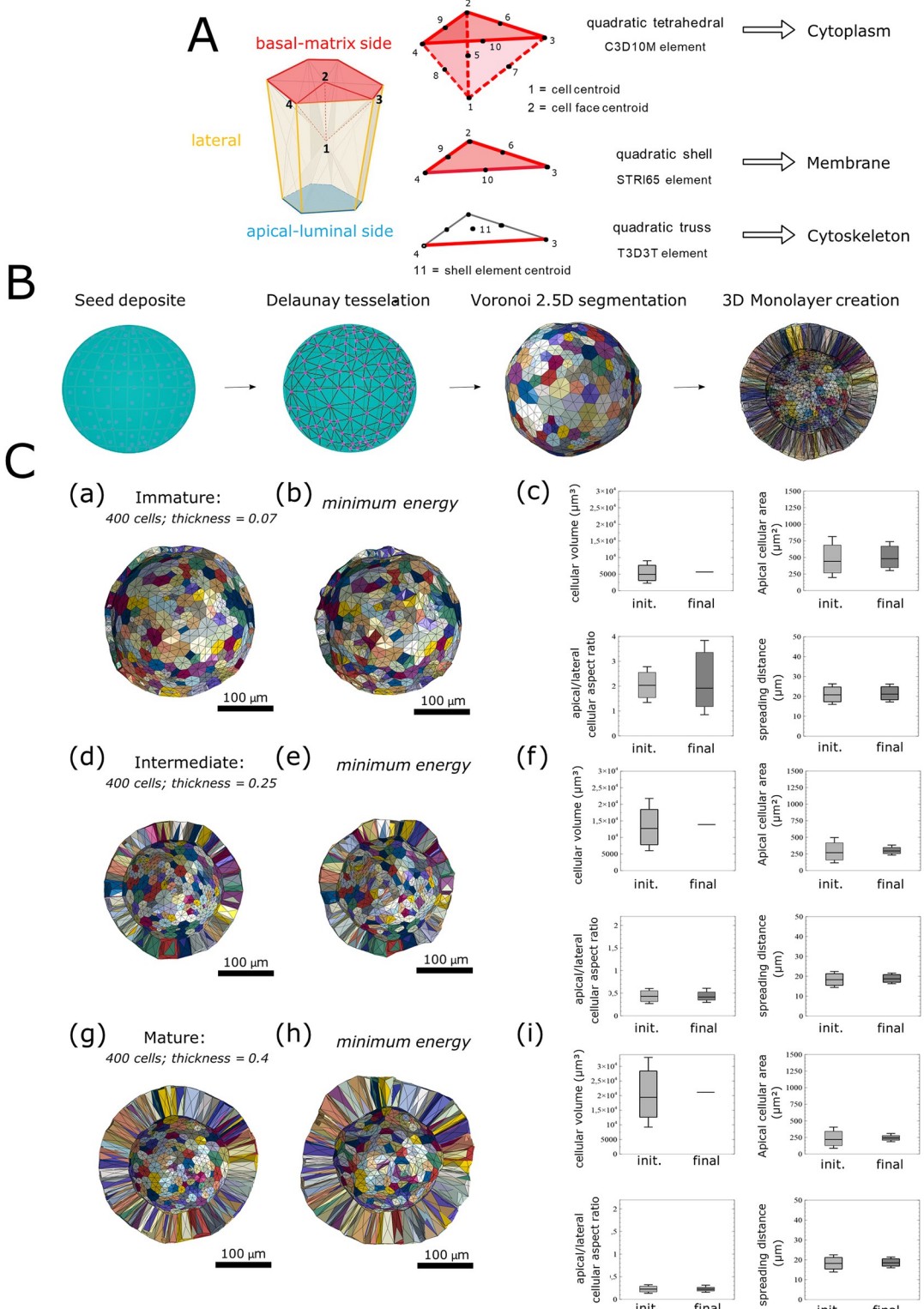

**Fig 2. Generation of the organoid active vertex model.** A: Description of the subcellular element model. The cell is composed of three element types: quadratic tetrahedral elements for cytoplasm volume discretization with C3D10M a predefined Abaqus element, quadratic shell elements for membrane discretization with a STRI65 predefined Abaqus element, and quadratic truss for cortex discretization with a T3D3T predefined Abaqus element contouring the apical-luminal (blue), basal-matrix (red), and lateral (green) surfaces of the cells to draw a "ring-like" structure. B: In silico organoids are generated

after four successive steps: seeds are distributed with a normal distribution function at the surface of a normalized sphere. A Delaunay 3D step creates a convex hull of the surface, and a Voronoi 3D step segments distinct cellular areas. The Voronoi is projected by dilation of the sphere with a defined "thickness ratio" relative to the radius of the final organoid (in μm). C: A previously published 3D AVM can be used to equilibrate our three "immature" (a), "intermediate" (d), and "mature" (g) models of organoid shapes. The AVM is a rheological model using gradient descent energy minimization to calculate the in silico organoid equilibrium state (b, e, h) [39]. Organoid morphology descriptors presented in four different distribution box plots for cellular volume, apical surface area, apical/lateral aspect ratio, and spreading distance distribution for one in silico immature organoid (c), one in silico intermediate organoid (f), and one in silico mature organoid (i), before (init.) and after (final) AVM solver resolution. Box and whisker plots indicate quartiles.

We generalized and applied this problem to the 3D model [41,42]. The mathematical formulation of this 3D AVM [43] can evolve to include a volume elastic coefficient for cell compressibility, cell face elastic coefficient, and lumen compressibility (Eq (1)).

$$E = \sum_e \Lambda_e \ell_e + \sum_f T_f A_f + \sum_e \left( \frac{K_A}{2} \left( A_c - A_c^0 \right)^2 + \frac{K_V}{2} \left( V_c - V_c^0 \right)^2 \right) + \frac{K_{lum}}{2} \left( V_{lum} - V_{lum}^0 \right)^2 \quad (1)$$

where $\Lambda_e$ is the line tension coefficient; $l_e$ is the edge length; $T_f$ is the surface tension coefficient; $A_f$ is the face area; and $K_A$, $K_V$, and $K_{lum}$ are the elasticity coefficients for the global cellular area ($A_c$), cellular volume ($V_c$), and luminal volume ($V_{lum}$), respectively. Index 0 corresponds to the initial reference state.

Minimum energy equilibrium is achieved by a gradient descent strategy using the constrained minimization algorithms of Broyden-Fletcher-Goldfarb-Shanno. We solved the system by adapting mechanical parameters (Table 2), which were determined to match the observed biological morphologies of the organoids as closely as possible (Fig 1). This process led to deformation of the vertex organoids (from Fig 2Ca, 2Cd, 2Cc to Fig 2Cb, 2Ce and 2Ch) *via* vertex displacement. We were then able to obtain information on individual cells' volumes, apical and basal areas, lateral areas, and aspect ratios due to the evolution of organoid morphology (Fig 2Cc, 2Cf and 2Ci).

Comparison of the data from the biological model (Fig 1F) and the VM (Fig 2C) indicated that this approach was neither effective nor adequate. After solving Eq (1) in the VMs, we observed a uniformity in the cell volume distribution and the cell area not representative of those observed at the different stages of the biological organoid model.

## Finite element method incrementation on a vertex organoid model

**Conceptual approach.** The discrepancy between models can be explained by the lack of consideration of the physical constraints experienced by the biological model in this modeling approach. To take these constraints (osmotic pressure [18], external solicitation [19,20], and active contraction [21]) into account, we used the 3D VM approach to generate an in silico

**Table 2. Active vertex model parameters: definitions of the parameters used for organoid architecture equilibrium resolution with Tyssue quasi-static solver (AVM simulation).**

| Parameter | Type | Value | Unit |
|---|---|---|---|
| Line tension | Edge | 50 | fJ/μm |
| Surface tension | Face | 10 | fJ/μm$^2$ |
| Contractility | Face | 0,1 | nNμm |
| Area elasticity | Cell | 1000 | fJ/μm$^4$ |
| Volume elasticity | Cell | 1000 | fJ/μm$^9$ |
| Volume elasticity | Lumen | 0,001 × cell number | fJ/μm$^9$ |

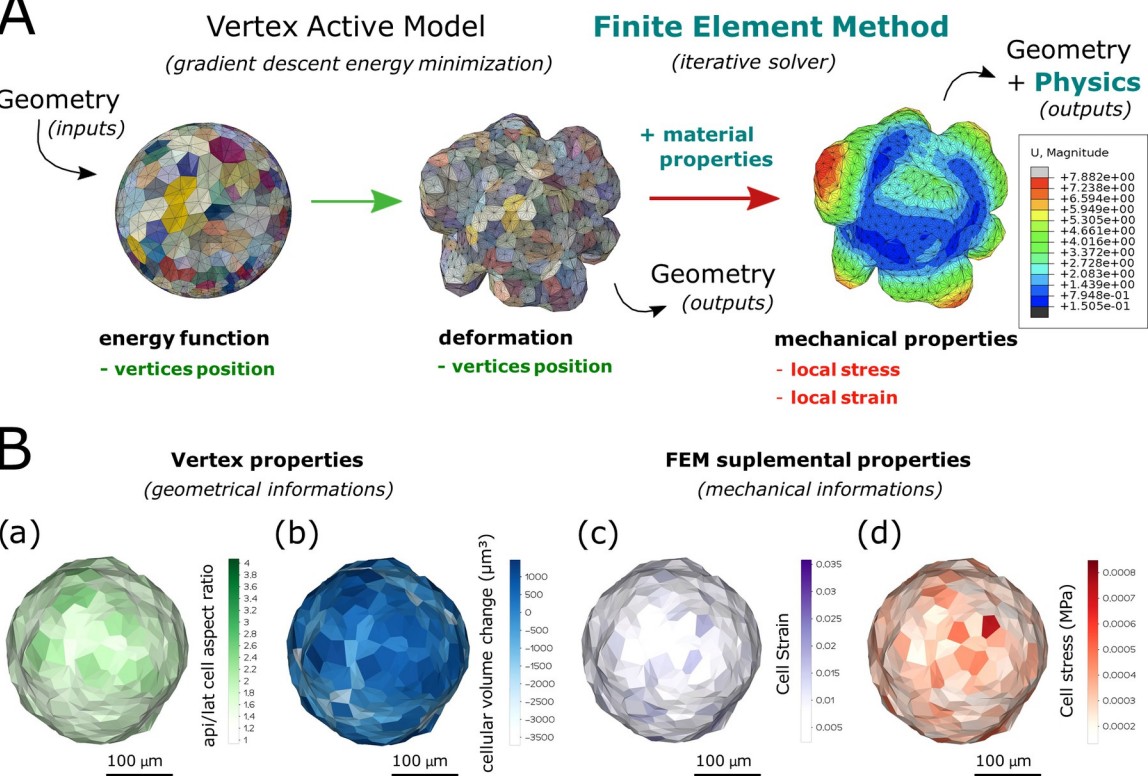

**Fig 3. Finite element method incrementation on an active vertex model of an organoid.** A: Schematization of implementation of the FEM on the deformation of a 430 cell in silico organoid resolved with a 3D AVM. FEM analysis and material properties are added to study the geometric and physics relationship in organoid architecture and examine the mechanical properties. B: In addition to geometrical data such as the apical/lateral area cell aspect ratio (A-L cell aspect ratio, (a)) or the differential volume after *vs.* before deformation (cellular volume change (b)), that are intrinsic properties of the mesh and accessible in all VMs, the use of FEM allows a mean strain value (c), and a mean cell stress (MISES) (d) to be computed for each organoid cell. Each property is measured at the element unit and averaged at the cellular unit for all 430 in silico organoid cells represented.

organoid (S1 Fig). In addition, to investigate the mechanical properties, we incorporated the material properties and stress information with respect to the deformation of the VM *via* the FEM on the VM.

First, we considered that the equilibrium state of organoid morphology depends on both the material properties associated with the object to be simulated and the constraints present in the system [43–49]. Thus, information regarding the constraints could be extracted by inducing a deformation to the VM in which the material properties have been applied (Table 2). Second, using an inverse approach, FEM allowed us to test the relevance of the material properties/stress pair to obtain an equilibrium state of the simulated model (Fig 3A).

The advantage of applying FEM to the VM was that we were able to extract both cell stress and cell strain from the in silico organoid (Fig 3B). Regarding cell stress, FEM allowed us to verify the von Mises [50] yield criterion (MISES), also known as "equivalent tensile stress," in the post processor by investigating the resultant stress for each element after deformation. Here, we used MISES to assess the stress level for each individual cell, thereby defining cell stress. Additionally, the cell strain could be extracted as the mean value of the strain of each element constituting the analyzed cell, itself computed through FEM.

**Modeling organoid deformation.** Our model analyzes living material according to the interaction between tissue and cell deformations, thereby informing a multi-scale model encompassing both individual cell shapes and the global organoid structure. We observed a non-uniform curvature distribution along the organoid shell, which can be classified into three types of alterations (Fig 4A): the first scenario displays the curvature of the tissue without any heterogeneity in cell shape or thickness (Fig 4Aa); the second displays decreased cell thickness at a specific location (Fig 4Ab); and the third exhibits an increased thickness of the epithelial monolayer along with a local increase in cell size (Fig 4Ac).

To generate the in silico model, we first considered a perfectly spherical initial state (Fig 4Bc). Using the FEA solver Abaqus, we decomposed the shape changes by implementing three types of stresses/loads to obtain elementary morphological deformations in the in silico model, generated on the basis of material properties (Table 3). From a mechanical viewpoint, the term "load" is used to express the stress imposed on the biological model.

The first load corresponds to an external solicitation [19,20] resulting in displacement of the surface node on the basal side of the cell (Fig 4Ba). External cell forces, defined by displacement at the node, are a frequently used method for calculating the resultant strain and constraint with FEM. We used this capability to model epithelial invagination in our FEM organoid. To mimic this local deformation, we applied a displacement field on the basal cell surface with an inverted Poisson's law distribution. Application of this positive pressure external to the organoid led to an invagination-like deformation, with a curvature of the basal surface accompanied by deformation of the apical surface (Fig 4Cb). Because measurement of the curvature is itself relative to the scale of the analyzed object, we used the apical/lateral and basal/lateral cell aspect ratios as curvature descriptors to quantify the regional deformation of the organoid on a cellular scale. Compared with the initial state, external solicitation slightly affected the cellular volume and apical/lateral aspect ratio, but had no effects on the apical cellular area or the spreading distance (Fig 4Ce).

The second load, active contraction [21], involves contraction *vs*. relaxation of the cortical elements (Fig 4Bb). Therefore, forces at the top surface of the cell (apical-luminal side) are associated with the need for apical contraction during epithelial invagination, whereas forces at the bottom of the cell (basal-matrix side) are more commonly associated with cellular expansion in response to disassembly of the basal actin cytoskeleton [51]. We decided to model cortical contraction by using an FEM model of volume expansion, which is frequently used to model contraction or thermal expansion of mechanical structures. In this context, contraction depends on two related material parameters: the expansion rate and specific heat. The expansion rate is the ratio between the variation in elementary volume and the variation in elementary load. From a mechanical perspective, specific heat corresponds to the amount of energy required to increase the elementary volume by one unit of load. In our context, it corresponds to the ability of our biological structure to inflate in response to an external load. Thus, simulation of the differential contraction of apical and basal cell surfaces was conducted by defining two expansion rates and a temperature condition that provokes deformation (Table 3). Cell expansion rates are associated with the state of the actin cytoskeleton. We mimicked this interplay by setting a higher expansion rate at the apical surface than the basal surface. A slight load can be considered a cell expansion parameter to induce an invagination-like deformation of the organoid (Fig 4Cc). Consequently, active contraction decreases cellular volume and the apical/lateral cellular aspect ratio (Fig 4Ce). No clear changes in the apical cellular area or spreading distance were observed.

The third load, osmotic pressure [18], coincides with the cytoplasmic pressures exerted on cell surfaces, from the cell's basal surface to its apical surface (Fig 4Bc). Whereas cytoplasmic pressure is a function of actomyosin contractility and water flux, independent control of the

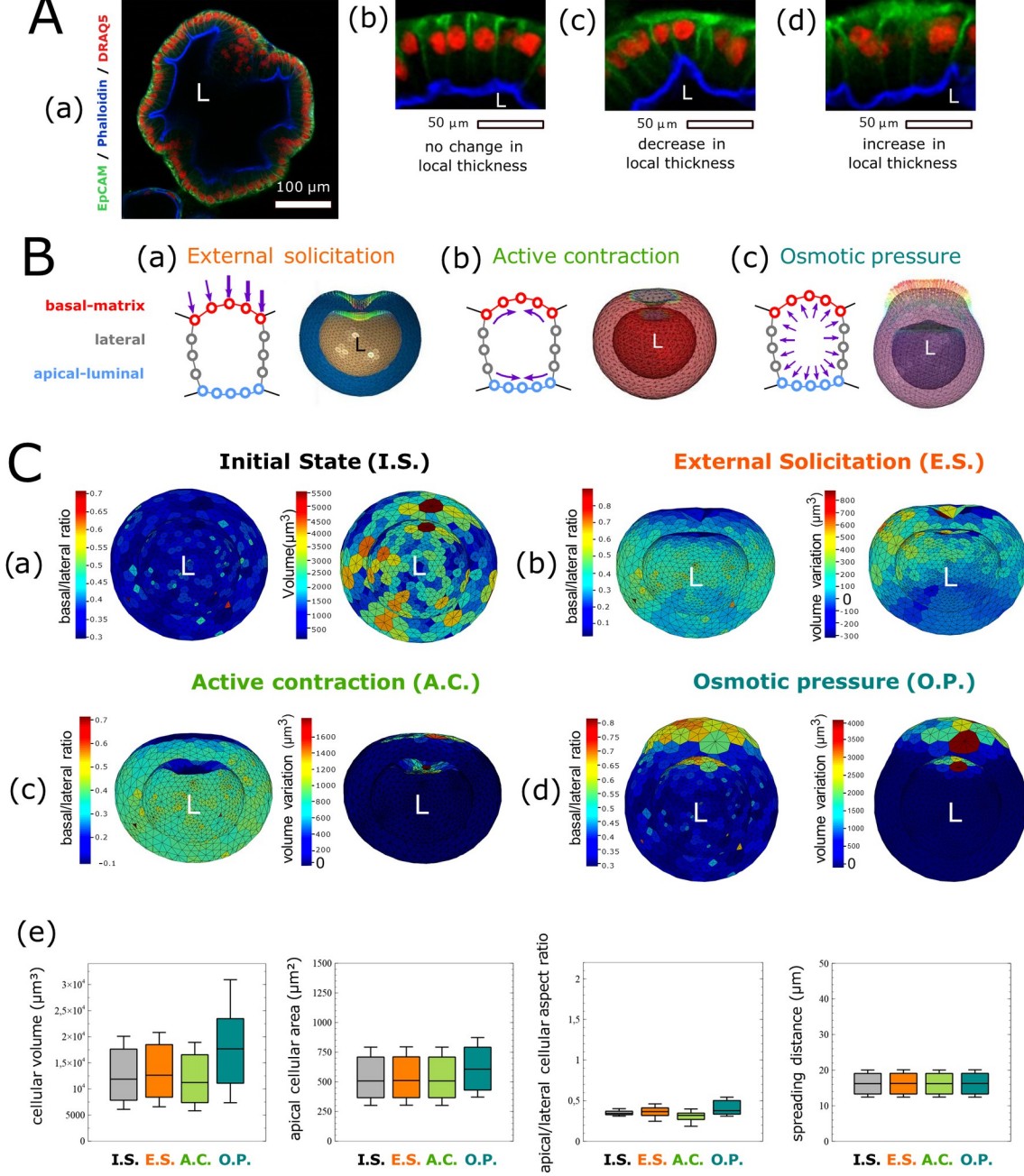

**Fig 4. Modeling organoid deformation.** A: z section of a fixed columnar organoid with phalloidin (blue) staining of actin filaments, EpCAM (green) immunostaining of epithelial cell adhesion molecules, and Draq5 (red) nuclear staining (a). (b-d) Three zoomed-in snapshots of the same fixed organoid, presenting three distinct cases of deformation with no negative or positive change in the local thickness of the epithelial monolayer. B: Schema presenting the three different types of controlled loads tested in the FEM model. (a) External solicitation controlling node displacement. Here, a gradual push on the apical surface results in an invagination phenotype. (b) Active contraction of both apical and basal surfaces defining a negative relative temperature, resulting in an invagination phenotype. (c) Osmotic pressure applied at the membrane elements' surface, with positive pressure dilating the cells and driving greater local thickness of the tissue. C: Sensibility of the 3D in silico organoid FEM model through controlled load constraints. Heat map 3D plot of two geometrical data types: basal/lateral area cell aspect ratio and volume at the surfaces of the cells for the common initial state (a'), a case of external solicitation (b'), a case of active contraction (c'), and one case of osmotic pressure (d'). The 3D representation was generated with Vedo [73] (Python). Organoid morphology descriptors resumed with four different distribution box plots for the cellular volume, the apical surface area, the apical/lateral aspect ratio, and spreading distance distribution for one in silico organoid, before (I.S.) and after the different types of load (e). Box and whisker plots indicate quartiles.

**Table 3. Material parameters used in the in silico organoid FEM model obtained by manual optimization based on both relevant global organoid morphology simulations and the literature.**

| Material | Element type | Young modulus (MPa) | Poisson ratio | Mass density (kton/m³) | Expansion | Specific heat | Viscous modulus | Relaxation rate |
|---|---|---|---|---|---|---|---|---|
| Cytoplasm | C3D10M | $2,10E^{-03}$ | 0,39 | $7,80E^{-09}$ | | | 0,01 | 0,3 |
| Apical membrane | STRI65 | $1,00E^{-03}$ | 0,35 | $7,80E^{-09}$ | | | 0,01 | 0,3 |
| Basal membrane | STRI65 | 2,00 | 0,38 | $7,80E^{-09}$ | | | 0,01 | 0,3 |
| Lateral membrane | STRI65 | $1,00E^{-05}$ | 0,42 | $7,80E^{-09}$ | | | 0,01 | 0,3 |
| Apical cortex | S3RT | $1,00E^{-03}$ | 0,35 | $7,80E^{-09}$ | $3,00E^{-02}$ | 200 | 0,01 | 0,3 |
| Basal cortex | S3RT | 2,00 | 0,38 | $7,80E^{-09}$ | $1,50E^{-05}$ | 400 | 0,01 | 0,3 |
| Lateral cortex | S3RT | $1,00E^{-04}$ | 0,42 | $7,80E^{-09}$ | $6,00E^{-08}$ | 400 | 0,01 | 0,3 |
| Lumen | C3D4 | $4,00E^{-06}$ | 0,25 | $1,00E^{-09}$ | | | | |

latter *via* regulation of osmotic forces is another important component emerging from mechanical models [52]. In our model, we considered cell cytoplasm to be viscoelastic [49,53,54] (Table 3). We therefore defined the Young modulus and a Poisson ratio associated with cellular compressibility. Using FEM, we tested osmotic pressure changes by applying pressure forces directly to specific cell surfaces as loading parameters. Osmotic pressure induced changes in cellular volume, the apical cellular area, and the apical/lateral cellular aspect ratio, but had no effect on spreading distance (Fig 4Ce).

In conclusion, our FEM model enabled us to model the three main elementary loading parameters and types of deformation that can affect tissue architecture. The approach allowed us to validate the material parameters describing elasticity and volume expansion in our model. Moreover, for each parameter, we identified the load ranges that must be introduced into the in silico model to simulate the morphological alterations occurring during the evolution of organoid morphology.

## Model validation

After working at the cellular level, we tested the approach on our global in silico organoid. Because biological organoids are grown in Matrigel matrix, we considered that the external solicitation [19,20] would be equivalent at every point around the organoid and therefore negligible in the in silico model. Nevertheless, external solicitation could easily be added by applying a single scaling factor to all different loads. To simplify the modeling approach, we neglected the external load in the global in silico model of the organoid. At the cellular or organoid level, active contraction depends on the concentration and activity of the actin cytoskeleton. Of these two parameters, activity is responsible primarily for altering the morphology [55], even if concentration and cortex architecture also influence cell shape [56,57]. With the same objective of simplifying the modeling process, we defined a homogeneous actin concentration by using an expansion coefficient, whereas inhomogeneous actin activity was represented by a load influenced by temperature in the in silico model. One feature of our approach is that cell contractility, specific to the activity of a biological system, can be finely controlled in the FEM model by using a specific load already described in the Abaqus software mechanical behavior laws. In the context of biological organoid models, two entities can reasonably be assumed to have semi-permeable membranes with respect to osmotic pressure. On the one hand, the cell membrane defines intracellular osmolarity. On the other hand, the closed epithelial monolayer defines intraluminal osmolarity. Thus, the contribution of these two osmolarities can explain the differential pressure present at the basal-matrix and apical-luminal surfaces of the organoid's global epithelial monolayer. This aspect is modeled in our in silico

model by the two different surface pressures required to obtain a global deformation representative of that observed in the biological model (Fig 1C).

On the basis of these concepts, we modeled the three steps representing the immature, intermediate, and mature morphologies, to understand the mechanics driving dynamic morphological changes during the live transition from immature to mature morphology in the biological model (Fig 5A). We generated a virtual organoid comprising 390 cells, corresponding to the average number of cells observed in our biological organoid cultures (Table 1). In step 1, the immature morphology was modeled by imposition of an osmotic pressure of 90 Pa and -500 Pa on the apical and basal surfaces of the in silico organoid, respectively, and we created lateral contraction with an expansion load of -100 a.u. for the lateral cortex. In step 2, we relaxed contraction at the lateral cortex (0 a.u.) and the apical and basal membrane by increasing the actin cytoskeleton volume expansion load to 100 a.u. to mimic the decrease in overall size of the biological organoid and its lumen, as well as the swelling of the apical surface (Fig 1C). In addition, we concomitantly decreased osmotic pressure inside the lumen and at the apical and basal surfaces by -0.9 kPa and -1 kPa. Finally (step 3), regarding mature organoid morphology, we simulated increases in the cellular area and volume by stepping up the osmotic pressure inside the cells. Consequently, swelling was modeled by increasing the active contraction of the apical actin cytoskeleton (-300 a.u.), which was compensated by an expansion of the basal actin cytoskeleton (400 a.u.). To support the increase in osmotic pressure inside the cell, with osmotic pressures at the apical and basal surfaces of -1.85 kPa and -1.21 kPa, respectively, we then stepped up the relaxation of lateral tension by increasing the volume expansion load of the lateral actin cytoskeleton (100 a.u.).

Consequently, in step 1 (immature morphology, Fig 5Ba), cell stress is distributed non-uniformly among the cells within the global organoid morphology. The external view (upper panel) presenting stress at the basal membranes displays areas with higher stress, whereas the internal view (lower panel), corresponding to the apical membranes, shows overall lower stress. Moreover, in each view, we observed areas of higher stress associated with increased deformation. In step 2 (intermediate morphology, Fig 5Bb), the organoid is globally more relaxed than in step 1, owing to the decrease in osmotic pressure in the lumen. In the external view, no areas display cell stress or local deformation, whereas in the internal view, areas show higher stress associated with global organoid deformation. Finally, in step 3 (mature morphology, Fig 5Bc), the overall range of cell stress is higher than that in step 2 but equivalent to that in step 1. Here, high cell stress intensity is restricted to only several cells and accumulates primarily on their apical sides, correlating with the organoid's global deformation.

The localization of cell strain in step 1 (Fig 5Bd) correlates with the organoid's global deformation. In step 2 (Fig 5Be), the organoid is relaxed in both views and displays only a low level of cell strain. Finally, in step 3 (Fig 5Bf), whereas low cell strain is observed in the external view, the internal view shows that higher local cell strain is associated with greater global deformation of the organoid. Overall, cell strain is distributed non-uniformly in steps 1 and 2, but is strongly correlated with the cell stress pattern. In step 3 and, more specifically, in the internal views, cell stress and cell strain patterns are mutually correlated. These findings are consistent, because we first modulated organoid inflation by adjusting global pressure, whereas global organoid constriction was modeled by increasing apical contraction.

As described earlier, we obtained information on individual cells' volumes, apical and basal areas, lateral areas, and aspect ratios due to the evolution of the organoid morphology, all of which appear to be in the same range of values as those observed in the biological model (Fig 5C). Comparison of the results extracted here with those of the biological model indicated that applying FEM on our vertex allowed us to validate the choice of material properties

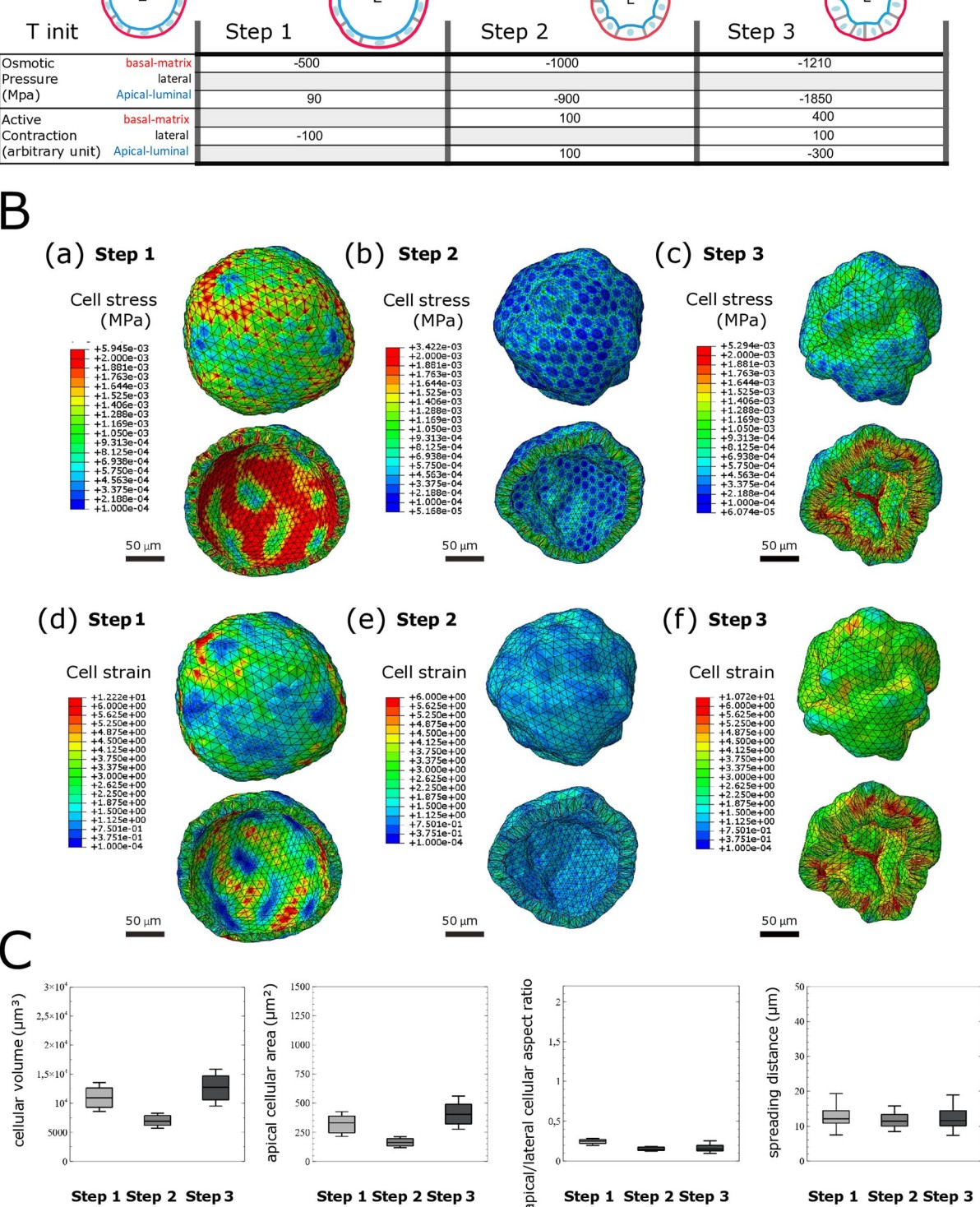

**Fig 5. Human intestinal organoid FEM model simulation of deformations occurring from immature morphology to mature morphology during a polarization event.** A: Schematization of the finite element analysis loading sequence used in the in silico model. Consecutive osmotic pressure and active contraction are applied to the apical (blue), basal (red), and lateral (gray) cellular surfaces of the in silico organoid model. B: Results of the FEM simulation on a 390-cell virtual organoid for step 1 (a, d), step 2 (b, e), and step 3 (c, f), equivalent to immature, intermediate, and mature organoid morphology, respectively. Stress (MISES) plot at subcellular resolution at the

global view (basal) and the cut view (apical) of the in silico organoid model (a, b, c). Strain deformation plot at subcellular resolution in the global view (basal) and the cut view (apical) of the in silico organoid model (d, e, f). C: Organoid morphology descriptors resumed with four different distribution box plots for the cellular volume, apical surface area, apical/lateral aspect ratio, and spreading distance distribution for one in silico organoid at step 1, step 2, and step 3. Box and whisker plots indicate quartiles.

describing the subcellular components. Thus, with this approach, we validated the mechanical behavior laws describing global viscoelasticity and volume expansion in our in silico model.

Although we were unable to perfectly reproduce the evolution of some cellular morphological properties (for example, the cellular volume and apical surface do not decrease monotonically), our approach helped us identify the range and types of loads (external solicitation [19,20], active contraction [21], and osmotic pressure [18]) involved in establishing the overall organoid morphology. We hypothesized that the adverse effects might be due to the application of a combination of several loads.

Moreover, our observations indicated that the level of deformation and the change in volume during human colon organoid morphogenesis (Fig 1) do not require the large deformation hypothesis. Therefore, we modeled deformation in a quasistatic simulation, using a linear viscoelastic constitutive relation for the tissue material. However, during morphogenesis, tissue may undergo large deformations, thereby implying a nonlinear regime. We consequently tested putative superelastic constitutive relations in our model.

Therefore, increasing the cell behavior law might potentially be feasible by incorporating additional physical phenomena, such as superelasticity or porosity (Table 4). As illustrated in Fig 6, these laws can be integrated into our model. However, determining the value of the constitutive parameters remains a challenge. Therefore, as the nonlinearity and the number of parameters in a behavior law increase, errors in estimating these parameters can have more pronounced effects on calculation accuracy. To ensure the model's relevance to its intended application, obtaining precise estimates of its parameters is crucial. In our case, the bulk modulus and void ratio for porosity, or the transformation strain for superelasticity, must be known. However, experimentally determining these parameters on a 3D human colon organoid model is highly complex and can lead to substantial uncertainty in the results.

As an example, Fig 6 and Table 4 demonstrate the potential incorporation of superelasticity (and porosity) into our model, using parameters adapted from the Abaqus documentation. In practice, we determined the parameters through trial and error, to ensure that the numerical convergence properties remained consistent with those of the elastic model. As indicated, doing so does not lead to model improvement.

Moreover, the porous-elastic law of behavior governed by the osmotic load can be used to model poroelastic effects. As shown in Fig 6, we integrated this law with the addition of porous elasticity parameters to the volumic elements constituting the cytoplasm of our virtual cells, as listed in Table 4. As described above, these parameters did not lead to better fit of either the individual cell morphology or the global organoid morphology of the synthetic organoid model *vs*. the biological model. Therefore, we decided not to include them in our modeling approach, and we focused only on the viscoelastic model, to minimize the potential errors.

On the basis of these observations, to validate our approach, we decided to test only the influence of one load, the osmotic pressure, given that this parameter is easily modulated in the biological model, because of Forskolin. Finally, we validated the biological relevance of our FEM-implemented VM through an inverse approach. We performed a swelling assay of organoids induced by Forskolin, a CFTR activator that drives uptake of ions and water into organoids, thereby equilibrating the osmotic pressure [58] (Fig 7). As expected, when comparing

**Table 4. Parameters used to test the porous elastic and superelastic states.**

| Material | Element type | Elastic | | | | |
|---|---|---|---|---|---|---|
| | | Young modulus (MPa) | Poisson ratio | Mass density (kton/m$^3$) | Expansion | Specific heat |
| Cytoplasm | C3D10M | | | 7,80E$^{-09}$ | | |
| Apical membrane | STRI65 | 0,013 | 0,33 | 7,80E$^{-09}$ | | |
| Basal membrane | STRI66 | 0,25 | 0,38 | 7,80E$^{-09}$ | | |
| Lateral membrane | STRI67 | 1,00E$^{-06}$ | 0,38 | 7,80E$^{-09}$ | | |
| Apical cortex | T3D3T | 0,014 | 0,31 | 7,80E$^{-09}$ | 0,03 | 200 |
| Basal cortex | T3D3T | 0,22 | 0,38 | 7,80E$^{-09}$ | 1,50E$^{-05}$ | 400 |
| Lateral cortex | T3D3T | 5,00E$^{-06}$ | 0,38 | 7,80E$^{-09}$ | 6,00E$^{-08}$ | 400 |
| Lumen | C3D4 | 4,00E$^{-06}$ | | 1,00E$^{-09}$ | | |

| Material | Element type | Porous elastic | | | |
|---|---|---|---|---|---|
| | | Log bulk modulus | Poisson ratio | Tensile limit | Void ratio |
| Cytoplasm | C3D10M | 0,02 | 0,33 | 0,015 | 0,6 |

| Material | Element type | Superelastic | | | |
|---|---|---|---|---|---|
| | | Young modulus (MPa) | Poisson ratio | Transformation strain | Start loading | End loading |
| Basal cortex | T3D3T | 0,014 | 0,31 | 0,057 | -0,005 | -0,009 |
| Lateral cortex | T3D3T | 0,22 | 0,38 | 0,057 | -0,005 | -0,009 |
| Lumen | C3D4 | 5,00E$^{-06}$ | 0,38 | 0,057 | -0,005 | -0,009 |

| | | Start unloading | End unloading | Start compression | Loading | Unloading |
|---|---|---|---|---|---|---|
| Basal cortex | T3D3T | -0,01 | -0,02 | -0,1 | 0,05 | 0,05 |
| Lateral cortex | T3D3T | -0,01 | -0,02 | -0,1 | 0,05 | 0,05 |
| Lumen | C3D4 | -0,01 | -0,02 | -0,1 | 0,05 | 0,05 |

organoids under Forskolin treatment *vs*. control conditions, we observed an increase in both overall organoid and individual cell volumes (Fig 7A).

To challenge our in silico model, we attempted to mimic the swelling observed on the biological organoids on our FEM-implemented in silico organoid model. We then generated a 320-cell FEM-implemented in silico model reproducing the parameters (cellular volume, area, aspect ratio, and spreading distance) extracted from the biological model to the greatest extent possible (Fig 7Ba, 7Bb, 7Bc; step 1-t0/step 1).

To model the effect of the swelling, we then modulated only the loads corresponding to osmotic pressure on the basal and apical surfaces (Fig 7Ba, 7Bb, 7Bc; T0+3.5 h/step2). In comparing the parameters of the biological models (Fig 7Ac) with those extracted from our in silico models (Fig 7Bf), we observed that the evolution of the organoid structures between the t0/step 1 and t0+3.5 h/step 2 stages resulted in similar global and cellular changes. FEM demonstrated that a threefold increase in osmotic pressure was sufficient to reproduce the swelling induced by Forskolin, but also revealed the evolution of cell stress among the organoid structures (Fig 7Bb/d and 7Bc/e). The correspondence among geometric descriptors (primarily cellular volume and apical cellular area) of the biological and in silico models provided an experimental validation of the biological relevance of our FEM-implemented vertex organoid model.

Thus, we identified and incorporated mechanics into our in silico FEM organoid model by using material properties. This model recapitulates the evolution of the organoid's overall morphology in a biologically relevant manner, while considering individual cell specificities. By considering the three main stresses/loads affecting tissue architecture and applying these stresses locally to the in silico tissue, we were able to identify the cellular mechanical properties involved in establishing organoid morphology.

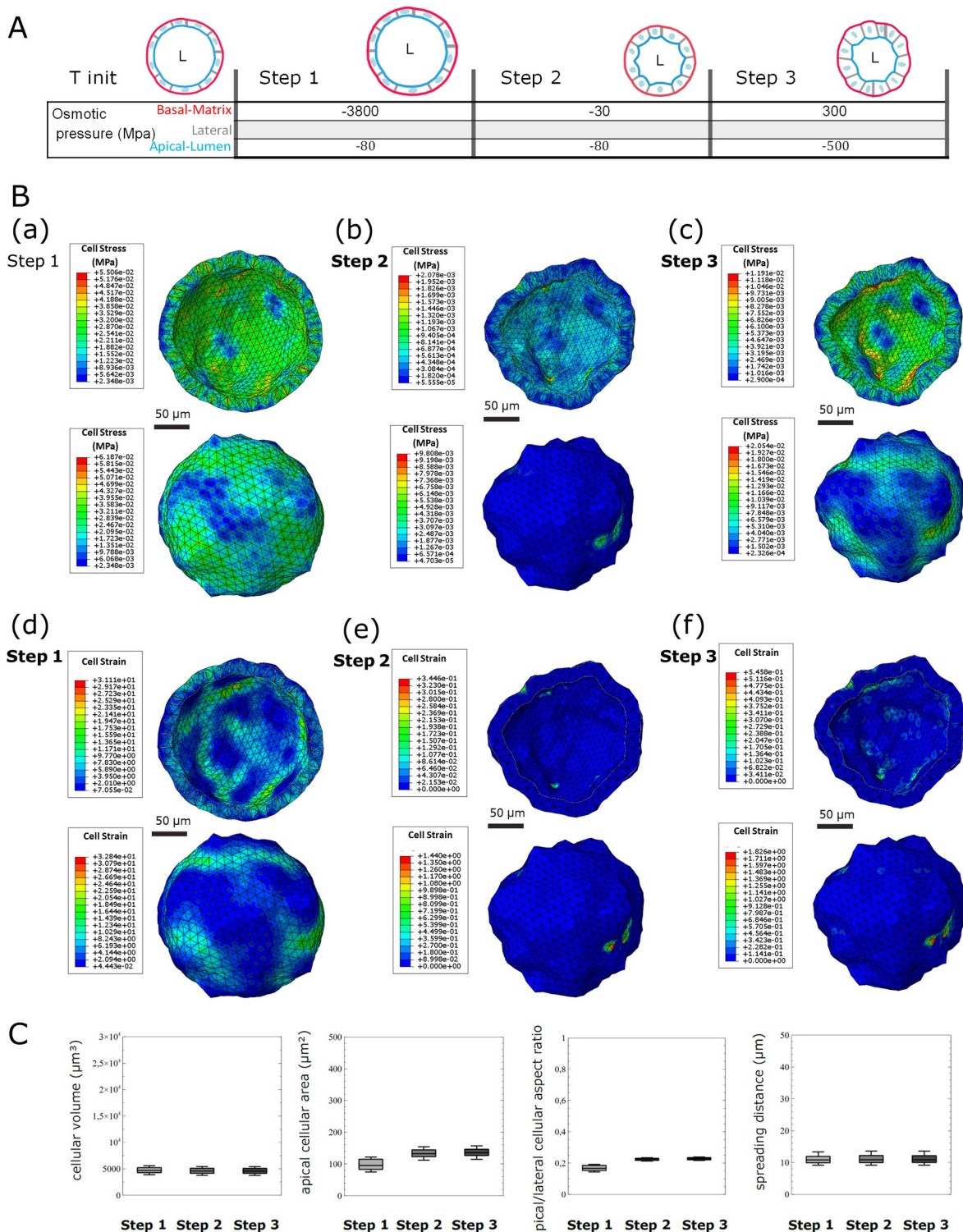

**Fig 6. Test of the porous elastic and superelastic states during deformations occurring from immature morphology to mature morphology during a polarization event.** A: Schematization of the finite element analysis loading sequence used in the in silico model. Osmotic pressure is applied to different cellular surfaces of the in silico organoid model, according to the parameters described in Table 4. B: Results of FEM simulation on a 390 cell virtual organoid for step 1 (d), step 2 (e), and step 3 (f), equivalent to immature, intermediate, and mature organoid morphology, respectively. Stress (MISES, a, b, c) and strain deformation (d, e, f) plots at subcellular resolution at the global view (basal) and the cut view (apical) of the in silico organoid model. C: Organoid morphology descriptors resumed with four different distribution box plots for the cellular volume, apical surface area, apical/lateral aspect ratio, and spreading distance distribution for one in silico organoid at step 1, step 2, and step 3. Box and whisker plots indicate quartiles.

## Discussion

The human colon organoid selected as a biological model to study the interplay between the biology and mechanics involved in establishing tissue architecture enabled us to extract cellular volumetric measurements through image analysis. We then calculated the morphological properties (cellular area, cellular volume, cellular aspect ratio, and spreading distance) characterizing this biological model. The data obtained for the immature, intermediate, and mature morphological stages were evaluated as potential quantitative geometrical descriptors of the tissue architecture evolution during the organoid's morphological evolution. These quantitative descriptors may serve as discriminant parameters for use in clustering analyses based on the study of tissue morphology through a machine learning approach.

Depending on their structure and use, three broad categories of *in silico* epithelium models have been proposed to study epithelial tissue behavior to date. Structured grid models, which are ideal for studying tissue self-organization with the diffusion of chemical gradients [57], model cells by repeating equivalent and uniform elements; therefore, accounting for variability in cell shape is difficult. In contrast, unstructured grid models, which do not use a predefined cell shape, are better suited to studying morphogenesis and have been extensively used in mechanical studies of 2D tissue organization [26,40,59–61]. Agent-based models use discrete tissue organization, with autonomous agents (spheres) representing cells, to decompose the overall problem, including cell migration, cell differentiation, or growth rate, into a sum of individual issues [41,61,62]. Although these models are extremely useful for tissue patterning, the discontinuity of the geometric structure makes them difficult to use in studies of mechanical properties [63].

Thus, to examine the biological and mechanical properties involved in the evolution of organoid morphology in a 3D multiscale approach from cell to tissue, we adapted an unstructured grid model, the 3D AVM, describing an epithelial monolayer theoretical biophysical model in 3D [17]. All epithelial VMs [26,64], whether 3D AVMs or 3D VMs, have shared structure and parentage with a vertex-based construct[40] defined as the junctions among at least three neighboring cells. Cell surfaces and volumes are then defined from the positions of the vertices, by taking the pressures and tensions specific to this geometry into account. The main similarity between the 3D AVM and 3D VM used is that 3D AVMs generally represent a more efficient numerical implementation enabling transition events for tissue rearrangement to be performed. Therefore, 3D AVM, which combines VM with active matter dynamics, is an evolution of 3D VM developed for cell monolayers and confluent epithelial tissues [24]. However, comparing the values of quantitative descriptors obtained with this adaptation *vs.* those extracted from the biological model indicates that the AVM is not sufficient (Fig 2). The discrepancy between biological and VMs can be explained because the physical stresses on the biological model are not considered or incremented in the VM. Thus, our strategy used the 3D VM approach to generate an in silico organoid. To investigate mechanical properties, we incorporated a FEM into this VM. For studying epithelial tissues, VM is widely used, because it represents a set of cells through points connected by segments [65].This point-based system serves as the foundation for minimizing an energy function, determining the equilibrium positions of the cell assembly. The 3D positions of the points are decision variables in an optimization problem, which, once solved, provides optimized positions. From these positions, structural and geometrical information, such as cell areas or equilibrium volumes, can be computed.

However, whereas the VM provides valuable geometrical information, it lacks the ability to measure mechanical properties. Another challenge, revealed in the AVM application (Fig 2), arises from the complex elastic response associated with target values of volume, area, or perimeter in AVM [66].

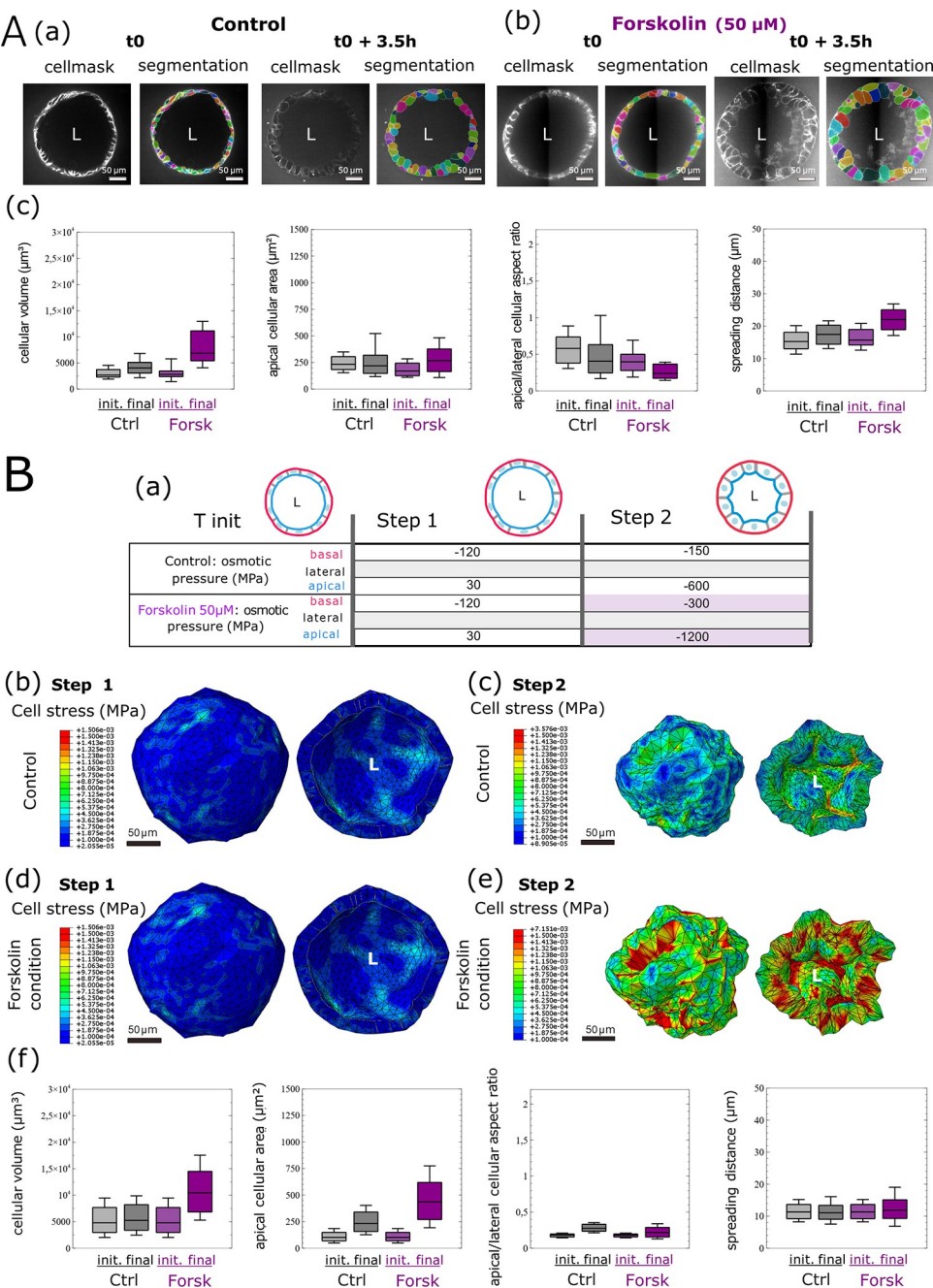

**Fig 7. Human intestinal organoid FEM model simulation of a swelling assay with Forskolin to increase osmolarity.** A: Quantification of organoid morphology during a swelling assay. On the basis of the CellMask live label imaged on confocal microscopy, 3D cellular compartments are segmented for a control organoid (a) or organoid treated with Forskolin at 50 μM concentration (b), before and 3.5 hours after treatment. Organoid morphology descriptors resumed with four different distribution box plots for the cellular volume, apical surface area, apical/lateral aspect ratio, and spreading distance distribution for one organoid at initial time and after 3.5 hours of culture *vs.* one organoid treated with Forskolin at 50 μM for 3.5 hours. Box and whisker plots indicate quartiles (c). B: Schematization of the finite element analysis loading sequence used for the in silico swelling assay (a). Consecutive osmotic pressure is applied to the apical (blue), basal (red), and lateral cellular surfaces of the in silico organoid model. Results of the FEM simulation on a 320-cell virtual organoid, presenting a stress (MISES) plot at subcellular resolution on a global or cut view of the in silico organoid model for step 1 (b, d) and step 2 (c, e), equivalent to the control (b, c) or swelling (d, e) condition, respectively. Organoid morphology descriptors resumed with four different distribution box plots for the cellular volume, apical surface area, apical/lateral aspect ratio, and spreading distance distribution for the in silico organoid swelling assay. Box and whisker plots indicate quartiles (f).

Our goal is not to invalidate the VM, which serves as a useful representation in configuration space and a first-step model for our approach, but rather to enhance it. By integrating FEM analysis, we can study various stress components, such as plane stress, shear stress, hydrostatic stress, and anisotropic stress associated with deformation.

This approach has high potential, because FEM enables refinement of the model's behavior without being hindered by computational complexity. The FEM enables material properties and stress information involved in the deformation to be considered and applied to the vertex organoid model. The main advantages of FEM are that it allows access to a finer level of simulation detail, incorporates complex mechanical behavior laws, and preserves coherence of the cell volume. The similarities observed between the biological model data and the FEM model data allowed us to validate the geometry of our model (including numbers of nodes, links, and interfaces), its mechanical properties (active contraction [21] and osmotic pressure [18]), and the loads imposed (external solicitation [19,20]), and thus the consistency of our modeling approach and its use. However, because our aim was to simplify the modeling approach for a monolayer epithelium, we decided to generate a continuous model using a continuous mesh and applying equivalent overall viscoelastic tension and pressure stress. The decision to simplify makes the constraint information inaccessible for each independent cell in our in silico model.

Several tools have been developed to measure mechanical stress in tissues at cellular and subcellular resolution [67]. Among them, two methods may be considered. FRET tension sensors consist of overexpressing "molecular springs" at the membrane. The spring's elasticity is known, and a fluorescent reporter is used to measure its elongation [68]. However, a major limitation of this approach with human organoids is the acquisition of 3D images and measurements on large 3D structures, in addition to differential membrane permeability that can create calibration challenges [69]. Another approach is monolayer stress microscopy, performed mainly on flattened cell cultures. This approach is currently among the most advanced for studying force deduction. However, it does not address all types of stress, notably osmotic pressure, thereby limiting studies of 3D organoid cultures with a lumen, because it cannot be used to investigate laminations or sphericity.

In contrast, our approach is very simple to use experimentally. Only the live imaging of organoid membranes is required to measure displacement and, owing to FEM analysis, all stress components (that is, plane stress, shear stress, hydrostatic stress, and anisotropic stress) associated with the deformation can be studied. Developing everyday tools allowing biomedical laboratories to measure numerous large organoids can be quite challenging, but recent advances in high content screening light sheet microscopy [68] and segmentation tools, such as Astec or Cellpose, are encouraging.

Our approach has substantial potential, because the use of FEM enables the mechanical behavior laws to be refined, without the limitation of computational complexity. However, greater complexity might be introduced by incorporating additional material properties already available in the Abaqus solver, such as hyperelasticity or porous elasticity. For instance, as demonstrated by Latorre et al., cells can exhibit superelastic behavior under osmotic stress [70]. Another idea is optimizing the problem with individual mechanical properties such as viscoelasticity for each subcellular component of each cell (cytoplasm, membrane, and cytoskeleton; Fig 2) to infer these local properties simply on the basis of the cellular shapes inside the organoid. The aim would be to study each cell individually, thereby allowing for more specific examination of the effects of mechanosensory signals on each cell, whether intrinsic or extrinsic. In fact, a reverse approach—starting from deformation observation, and moving toward stress and strain evaluation—could aid in assessing the evolution of material properties at the cellular level within epithelial tissues. Future endeavors of potential interest may include

evolving our model by coupling the VM formulation with other methods, such as deformable cell models, which have recently been demonstrated to be promising tools for creating virtual organoids [71].

Our innovative FEM model takes advantage of the accurate description of each cell and specific mechanical laws associated with cell behavior to model the global tissue. This multi-scale model offers a promising tool for understanding global tissue alterations and possible contributions to individual cellular events. Challenging this FEM approach to model cell specification during morphogenesis might prove intriguing, particularly in systems such as vertebrate neural tubes or Drosophila oocytes, in which morphogen gradients and cell specification are well characterized [72]. However, the roles of individual cell mechanics in these processes remain to be determined. Another suitable biological model is the ascidian, which offers several advantages, including a small number of cells, excellent imaging capabilities, and a well-documented process of cell lineage acquisition [73,74]. However, when focusing specifically on the human colon, the human colon organoid model cultured in Matrigel is not ideal for studying how chemical gradients (morphogens) regulate cell shape. This limitation is because, under the current experimental setup, the environmental chemical conditions are uniform around the organoid, preventing the formation of specific gradients. Therefore, in the future, conducting similar studies on a human colon organoid culture established in a gut-on-chip system would be valuable. This setup would better mimic the mechanical and chemical environmental conditions of human tissue in vivo, and would allow for the reproduction of cell compartmentalization and chemical gradients across different cell types.

From biological and medical perspectives, the ability to link individual cell behavior to global tissue alteration might find future applications in image-based diagnostic and screening approaches. As argued by Tzer et al., correlating such in silico models with biochemical/biological (single-cell RNA sequencing) and imaging (segmentation and feature extraction) data can advance understanding of the establishment and regulation of tissues by mapping the combination of such information within the tissue structure [75]. Here, for instance, the inclusion of our in silico FEM model in novel AI driven automatic processes might help identify very early tissue alterations in patient colon organoids and enable more personalized follow-up.

## Materials and methods

### Ethics statement

Colon samples were obtained from biopsies of patients undergoing endoscopy at Toulouse University Hospital. Patients provided oral informed consent and were included in the registered BioDIGE protocol, approved by the national ethics committee (NCT02874365) and financially supported by Toulouse University Hospital.

### Organoid culture

Colorectal crypt isolation was performed as previously described [34]. Fresh Matrigel (Corning, 356255) was added to isolated crypts. A 25 μL volume of Matrigel containing 50 crypts was plated in each well of a pre-warmed eight-well chamber (Ibidi, 80841). After 3 days of culture, organoid stocks were frozen. From these stocks, organoids were later thawed, plated in Matrigel, and expanded for amplification under previously described culture conditions [34]. After 10 days of culture, the medium was replaced with 250 μL Intesticult ODM human basal medium (StemCell technologies, 100–0214), thereby allowing for the generation of intestinal organoid cultures with physiological representations of the stem cell and differentiated

populations. The cultures were followed for the times indicated in the figures. For swelling assays, the cultures were treated with Forskolin (50 μM) or control (DMSO, 1.25%) and imaged just before drug addition (t0) and 3.5 h after treatment (t0+3.5 h).

## Immunostaining and live imaging

Organoids were fixed with PBS solution containing 3.7% of formaldehyde solution for 5 min at 37˚C. The organoids were then washed in PBS and permeabilized with a PBS solution containing 0.5% Triton X-100 for 20 min at 37˚C. DNA was subsequently stained with Draq5 (Ozyme, 424101, 2 μM) for 30 min, and membranes were stained with anti-EpCAM antibody (Cell Signaling Technology, VU-1D9), whereas the actin network was stained with Alexa Fluor 594 phalloidin (Thermo Fisher, A12381, 40 nM). Finally, glass coverslips with Matrigel domes were immersed in PBS for imaging. For live experiments, the cytoskeleton was stained with SPY650-FastAct or BioTracker-488 microtubule dye (SCT142, Sigma-Aldrich), and DNA was stained with the SPY555-DNA probe (SpyDNA). All fluorescent live cell probes came from Spirochrome and were used at a final 1/500 dilution 4 hours before the start of live imaging. Image acquisition was performed with a two-photon microscope (Bruker 2P+, 20× diving lens objective) for fixed experiments and an Opera Phenix HCS microscope (Perkin Elmer, 40× objective) for live experiments. The images were analyzed in FiJi ImageJ software [76].

## Image analysis

Cell segmentation was performed with the Cellpose2 machine learning-based 2D approach, adapted for each individual organoid analyzed, and 2D segmentations were subsequently stitched in 3D with the same software [35]. Luminal volume was segmented manually with Napari [77] software. The 3D isometric interpolation was performed with VT [78] Python libraries for segmented images (labels) or Napari for raw data, and manual corrections of the segmentation were performed with Morphonet [79] or Napari [77] to select and correct only the cells entirely present in the 3D acquired field, to achieve accurate estimation of the cellular interfaces. The number of cells selected for the analysis is presented in Table 1.

## Model implementation and data analysis

We used the Python (3.7.11) libraries NUMPY and SCIPY to generate the mesh staked in Pandas datasets that could be exported in HDF5, OBJ (for surfaces), or VTK formats with MESHIO. The Python code used primarily the TYSSUE [43] library for monolayer creation. For remeshing and properties extraction, the CGAL and VT [78] libraries containing image-oriented algorithms written in C were used, as well as VTK or Vedo [80] for visualization. Conversion into the Abaqus (Dassault Systems) input file was coded in Python to create the final in silico organoid.

## Finite element formulation

We selected a collection of elements useful for our model and preconfigured in Abaqus. Each type of element was specific to the subcellular unit to be meshed.

**Table 5. Range of values used in the preliminary experimental design, and resultant set of parameters applied to our model.**

| Values | Bulk modulus | Poisson ratio | Tensile limit | Void ratio |
|---|---|---|---|---|
| Minimum | 0.005 | 0.27 | 0.012 | 0.55 |
| Maximum | 0.02 | 0.35 | 0.017 | 0.65 |
| Calculated | 0.0209 | 0.3299 | 0.0145 | 0.6 |

- Cytoskeleton: Quadratic C3D10M Abaqus tetrahedral element connecting the centroid of the cell and part of the cell face. Tetrahedral elements were chosen because they are more convenient than hexahedral elements to cover a sphere [81] uniformly and are more flexible for complex cell shapes [82].

- Membranes: Quadratic STRI65 Abaqus shell elements connecting only the nodes of the preceding volumic elements present at the surfaces of the cells.

- Actin cytoskeleton: The active network present in the intracellular space close to the membrane was modeled by sharing the node of the membrane with first order T3D3T Abaqus thermal truss elements.

## Material model parameters

We defined the physical behavior of the FEM organoid in four distinct domains—cell cytoplasm, cell membrane, cell cortex, and the extracellular lumen—each with specific elements, i.e., specific mechanical properties (Table 3). For simplicity, we assumed that the cytoplasm is viscoelastic [49,53,54] and that this elasticity is isotropic. In the literature, epithelial cell cytoplasm elasticity is characterized by a Young modulus ranging from 0.3 to 100 kPa, depending on the biological model and measurement method used [46,48,49,61,62]. Viscoelasticity is controlled by a stress relaxation modulus defined as equivalent for all cell components and, given the quasi-static resolution, time is irrelevant (Table 3) [83]. In all our models, we fixed an arbitrary value of 21 kPa for cytoplasm elasticity after validation of simple sheet deformations, associated with a Poisson ratio of 0.39. High Poisson ratios (near 0.5) are widely used for deformations, which occur at constant volume. Similarly, we gave the membrane elements elastic behavior to represent membrane elasticity, according to the literature. Given the differing nature of the interfaces, we assigned higher elasticity to the apical membrane than the basal membrane, which is surrounded by a basal lamina.

The last surface of the cell, the lateral membrane, at the interface between neighboring cells, specifically connects two cellular membranes. In our continuous VM, to simplify the problem, we did not duplicate membranes for cell-cell contacts. With this unique interface, we decreased the number of contacts to attribute and compute when solving iterative operations. The constraints at this lateral membrane interface were treated with a low Young modulus of $4.5E^{-06}$ kPa. Consequently, constraint at cell-cell interfaces is due primarily to a pressure equilibrium exercise by cytoplasmic pressure.

## Optimization of material properties

First, we explored the minimum and maximum values allowing for a solution of the system in terms of computation, ensuring that Abaqus could find a convergent solution. The next step involved exploring combinations of these loading values to best represent the morphology observed in the initial stage. After validating the first step of the simulation, which involved inflation of the organoid, we proceeded with the second step of deflation, and finally the last step, which focused on enlargement of the organoid shell. This solution is not unique but is the best that we could achieve in terms of biological relevance for the overall morphology of the organoid. We applied the loads intuitively to simulate the process, on the basis of understanding of the basic mechanisms driving the biological events.

To obtain the best possible set of parameters, this problem should ideally be formulated as an inverse problem and solved with numerical optimization algorithms. To set up this inverse problem, we would need time-series data enabling displacement realignment and correlation. However, the non-linearity resulting from the combination of porous elasticity, viscoelasticity, and superelasticity makes this calculation too complex.

Tests of arbitrarily chosen combinations of parameters also lead to majority of non-convergence of the calculation. Because the parameter definition domain is highly irregular, we were unable to perform recalculation of parameters with the conventional optimization method. For illustration, the complete experiment plan with the four parameters at two levels indicated in the table below was conducted. Only six combinations allowed for the convergence of calculation and consequently yielded a usable result. We therefore chose to manually optimize the parameters, guided by the first results of this experiment and approximately ten other complementary trials. The values chosen are derived from manual optimization in the vicinity of the best set of parameters obtained initially (Table 5).

We optimized the model to the best extent possible with the calculated mechanical parameters, but we did not find another solution to the porous elastic problem that produced better results in terms of the global organoid morphology.

## Model loading parameters

To control deformation in our in silico organoid FEM model, we used the following three types of loads.

*External solicitation* [19,20] was modeled with node displacement on the basal membrane and could also be used to study displacement extracted from tracking information given by live analysis.

*Active contraction* [21] was modeled with the volume expansion method in Abaqus. This process requires a temperature-displacement solver based on specific material properties, a thermal expansion coefficient, and specific heat, in addition to the definition of both initial and final temperature as a boundary condition. The load for active contraction is used as a relative temperature that is decorrelated from the actual temperature set in our biological system.

*Osmotic pressure* [18] is modeled with uniformly distributed pressure at the surfaces of the cells. We adjusted this pressure with vectors directed outside the cells and perpendicular to the element area.

## Supporting information

**S1 Fig. Schematic representation of the construction of our EF model compared with the vertex model.** The vertex model uses the vertices of the mesh as optimization variables, thereby minimizing the energy of the active 3D vertex model to obtain geometric information. The proposed method uses these vertices as the initial geometric configuration, then meshes them by using finite elements, and finally labels them to decipher the individual cells. This approach enables geometric and mechanical information regarding not only points but also volumes to be obtained.
(TIF)

## Acknowledgments

We thank all patients who agreed to provide their tissue for research purposes and made this study possible; Guillaume Gay, who gave us permission to use his Tyssue Python library; and Gaëlle Recher and Camille Douillet for their critical reading of the manuscript and suggestions.

## Author Contributions

**Conceptualization:** Julien Laussu, Stephane Segonds, Florian Bugarin, Audrey Ferrand.

**Data curation:** Julien Laussu, Deborah Michel, Léa Magne.

**Formal analysis:** Julien Laussu, Deborah Michel, Léa Magne, Steven Marguet, Vincent Velay.

**Funding acquisition:** Frederick Barreau, Florian Bugarin, Audrey Ferrand.

**Investigation:** Julien Laussu, Deborah Michel, Léa Magne, Dimitri Hamel, Muriel Quaranta-Nicaise.

**Methodology:** Julien Laussu, Deborah Michel, Léa Magne, Steven Marguet, Dimitri Hamel, Muriel Quaranta-Nicaise, Vincent Velay, Florian Bugarin, Audrey Ferrand.

**Project administration:** Stephane Segonds, Florian Bugarin, Audrey Ferrand.

**Resources:** Emmanuel Mas, Florian Bugarin, Audrey Ferrand.

**Software:** Julien Laussu.

**Supervision:** Florian Bugarin, Audrey Ferrand.

**Validation:** Julien Laussu, Stephane Segonds, Steven Marguet, Florian Bugarin, Audrey Ferrand.

**Visualization:** Julien Laussu, Deborah Michel, Léa Magne, Florian Bugarin, Audrey Ferrand.

**Writing – original draft:** Julien Laussu, Deborah Michel, Léa Magne, Stephane Segonds, Vincent Velay, Florian Bugarin, Audrey Ferrand.

**Writing – review & editing:** Julien Laussu, Stephane Segonds, Florian Bugarin, Audrey Ferrand.

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
