## [Decision Letter · Decision Letter 0]

21 Jun 2024

Dear Dr Ferrand,

Thank you very much for submitting your manuscript "Deciphering the interplay between biology and physics with finite element method-implemented vertex organoid model" for consideration at PLOS Computational Biology.

As with all papers reviewed by the journal, your manuscript was reviewed by members of the editorial board and by several independent reviewers. In light of the reviews (below this email), we would like to invite the resubmission of a significantly-revised version that takes into account the reviewers' comments.

We cannot make any decision about publication until we have seen the revised manuscript and your response to the reviewers' comments. Your revised manuscript is also likely to be sent to reviewers for further evaluation.

Sincerely,

Nir Gov

Academic Editor

PLOS Computational Biology

Pedro Mendes

Section Editor

PLOS Computational Biology

Reviewer's Responses to Questions

**Comments to the Authors:**

Reviewer #1: In this manuscript, the authors combine a vertex model with a finite element model of mechanical deformation to describe shape changes in organoid tissue (human colon epithelium).

The question of time-dependent 3D deformations of tissue is important in physical biology. The work makes good computational progress and is validated by experiments. It shows how the material deformations and constraints not captured by vertex modeling, can in fact be captured by including finite element description of material analysis.

However, I have significant concerns regarding the general applicability and biological relevance of the model, as well of intuitive arguments and quality of writing, for which I do not recommend publication in the present form.

1.One of the advantages of finite element modeling is that it can handle large deformations in complex geometries. However, this potential does not seem to have been completely exploited in the present work. Although not explicitly stated, the authors seem to be using a linear viscoelastic constitutive relation for the tissue material. Tissue undergoing large deformations during morphogenesis may not however remain in this linear regime. Can the authors consider hyperelastic constitutive relations in their model and does this show better agreement with experiments? In particular, it has been shown that epithelial cells are “superelastic” under osmotic stress (joint work by Arroyo and Trepat labs), that is they can stretch several times their original volume.

2.During osmotic expansion, or active contraction, cells regulate their volume through fluid flow. This should lead to poroelastic effects in their dynamics. Can the authors implement this in their FEM model?

3.How general and biologically significant are the kind of shape transitions demonstrated for this system? Does the immature to mature structural changes shown in Fig. 1C have some biological function, and is it generic in many kinds of epithelial tissue?

For example, It is known that in the Drosophila oocyte (see e.g. Brigaud et al. 2015 Biology Open), such squamous and columnar cell shapes can co-exist. Can the authors show such a possible co-existence in their model, as was indeed predicted in Ref. 32 that the authors cite? Based on the model of Ref. 32 (Hannezo et al) and experimental studies such as by Widman and Dahmann, Journal of Cell Science 2009, it was theoretically proposed by Dasbiswas, Hannezo and Gov, Biophysical J 2018, that chemical signaling gradients (“morphogens”) can influence cell mechanics and shape. A discussion of such mechano-chemical effects and how to incorporate them in the FEM framework can strengthen the manuscript.

Without a discussion of the biological relevance of the types of deformation described here, and the general applicability of this model to systems beyond the one case it has been tested on, the paper does not seem very complete.

4.Most experiments described seem to be on the wild type of the same epithelial tissue. Can the authors do knockdown or drug treatment experiments that will allow a more complete exploration of the parameter space in their model?

5.In figure 4C e, it seems strange that active contraction does not reduce the apical area relative to the initial state. Can the authors intuitively explain this observation?

Minor comments

1.The language of the manuscript needs improvement. There are several unclear sentences or unconventional usages in English that may make understanding difficult. For example, the use of “increment” in “incrementing a finite element model” in lines 69-70 is not familiar to me.

Sentence in 364 is grammatically incorrect (wrong word order). There are typos in lines 76, 305 and several other places too long to list. The authors should carefully proofread their manuscript at the revision stage.

2.It should be made clear at the outset that apical/basal surfaces correspond to the inner/outer surfaces in this geometry. This may not be clear to non specialists not familiar with the specific model system used here. I am seeing this mentioned first in line 286 of the manuscript, which is a bit too late.

3.Maybe mark the parameters in table 1, diameter and thickness ratio, in Fig. 1 to help the reader.

Reviewer #2: This paper presents a model of epithelial spheroid static deformation, implemented with a FEA method. The model includes terms for volumetric, surface, and line forces/stresses/force densities. Some example simulations are presented.

The paper is poorly composed, so that it is difficult to determine exactly what the method is. The introduction gives a lot of motivation, but no roadmap for what is done in this study. The conclusions are overblown.

Tables of parameters are presented without justification, and in some cases are presented without appropriate units. It is not clear whether there have been any tests of model sensitivity (which there should be). Some parameters are given in a.u. without explanation.

Much of the writing is geometrically sloppy, such as (line 66) "each cell is represented by polygons", which leaves the reader wondering what dimension is being referred to, and the confusing "all cells are attached to the basal and apical planes" (line 130).

There are also nonstandard terms, like "external solicitation", which do not contribute to the reader's understanding. The awkward phrasing and grammatical errors make me wonder if the MS was first written in French, and machine-translated without intervention from a native English speaker. On the other hand, there are awkwardly complex sentences such as "Deciphering the individual contributions of these intertwined cell events into the intestinal tissue architecture establishment necessitates producing a powerful computational model gathering biology, physics, and geometry" (line 58) which suggests AI writing. If the paper is resubmitted, I strongly urge the authors to use a human copy editor.

The figures need revision. Fonts are too small in many places, and inconsistent color scales are used for comparisons.

Overall, this paper presents a modeling methodology that has some potential, though there are recent papers which do much the same thing but with more accuracy and versatility (such as time dependence).

**Have the authors made all data and (if applicable) computational code underlying the findings in their manuscript fully available?**

Reviewer #1: **No: **The code could be shared as a github repository. Currently, it says "available upon reasonable request" which may not agree with the journal requirement.

Reviewer #2: **No: **"Data will be made available upon request" is the only relevant statement.

PLOS authors have the option to publish the peer review history of their article (what does this mean?). If published, this will include your full peer review and any attached files.

Reviewer #1: No

Reviewer #2: No
---

## [Decision Letter · Decision Letter 1]

27 Sep 2024

Dear Dr Ferrand,

Thank you very much for submitting your manuscript "Deciphering the interplay between biology and physics with a finite element method-implemented vertex organoid model" for consideration at PLOS Computational Biology.

As with all papers reviewed by the journal, your manuscript was reviewed by members of the editorial board and by several independent reviewers. In light of the reviews (below this email), we would like to invite the resubmission of a significantly-revised version that takes into account the reviewers' comments.

One important aspect mentioned by reviewers is access to source code, the PLOS Computational Biology software policy requires that the source code be available publicly, either in a public repository, or as supplementary material to the article. Please see the full policy at https://journals.plos.org/ploscompbiol/s/code-availability

We cannot make any decision about publication until we have seen the revised manuscript and your response to the reviewers' comments. Your revised manuscript is also likely to be sent to reviewers for further evaluation.

Sincerely,

Nir Gov

Academic Editor

PLOS Computational Biology

Pedro Mendes

Section Editor

PLOS Computational Biology

Reviewer's Responses to Questions

**Comments to the Authors:**

Reviewer #1: the review is uploaded as an attachment

Reviewer #3: The authors suggest a tool for mechanical analysis of the behavior of human colon epithelial cells on an organoid spherical shell. The tool integrates several published or commercial computational tools, including a classic vertex model, previously defined finite elements models in Abaqus, and other tools for image analysis and 3D multicellular displays.

I think this could be a valuable tool for the community, and in general, I concur with the suggested interpretations. However, there is a big problem with the way the tool is presented and motivated. The authors also fail to cite prior relevant publications, or poorly refer to them. Furthermore, I find the language and wording used in many parts throughout the MS to be unclear or awkward.

Another cardinal issue here is the lack of effort to make it clear that the suggested tool is an integration of many available tools (some commercially). It is written in the methods section and in some parts of the text, yet, it is not spelled out immediately from the beginning. If it will be so, I think, it will make this work more attractive and useful for others. Lastly, and regarding usefulness, the fact that the suggested tool is mainly an integration of many other available tools calls for full availability in the Supp Material or some other cloud service. This should include a detailed installation and instruction guide and an available example data set to run one of the analyses presented in the MS.

Further major comments:

-- Title: – promises something very general, yet the MS provides something much more specific. Please modify.

-- Abstract: (in a way similar to the title) I found it too general and lacking concrete observations and conclusions, not from the biophysics aspect nor from the computational aspect.

-- Line 66: the end of the sentence is unclear.

-- Line 69 and Line 76 (“Vertex Models”): It seems that many important papers on cell shape in development and in curved surfaces, specifically with regard to the vertex model and the concept of multicellular jamming, are not mentioned here. To name a few:

1. Bi, D., J.H. Lopez, J.M. Schwarz, and M.L. Manning, A density-independent rigidity transition in biological tissues. Nat Phys, 2015. 11(12): p. 1074-1079.

2. Bi, D., X. Yang, M.C. Marchetti, and M.L. Manning, Motility-Driven Glass and Jamming Transitions in Biological Tissues. Physical Review X, 2016. 6(2): p. 021011.

3. Atia, L., D. Bi, Y. Sharma, J.A. Mitchel, et al., Geometric constraints during epithelial jamming. Nature Physics, 2018.

4. Spurlin, J.W., M.J. Siedlik, B.A. Nerger, M.-F. Pang, et al., Mesenchymal proteases and tissue fluidity remodel the extracellular matrix during airway epithelial branching in the embryonic avian lung. Development, 2019. 146(16): p. dev175257.

5. Tang, W., A. Das, A.F. Pegoraro, Y.L. Han, et al., Collective curvature sensing and fluidity in three-dimensional multicellular systems. Nature Physics, 2022.

The list goes much further, and I would suggest modifying the introduction such that it will at least acknowledge and place in the proper context (of this paper) this body of literature.

-- Line 76, “volumetric vertex model”: The authors need to better survey similar tools that combined finite element approach and the vertex model such as

Smith, A.M., R.E. Baker, D. Kay, and P.K. Maini, Incorporating chemical signalling factors into cell-based models of growing epithelial tissues. Journal of Mathematical Biology, 2012. 65(3): p. 441-463.

And also survey other studies that proposed alternatives to address the vertex model limitations. For example deformable cell models (DCMs)-

Runser, S., R. Vetter, and D. Iber, SimuCell3D: three-dimensional simulation of tissue mechanics with cell polarization. Nature Computational Science, 2024. 4(4): p. 299-309.

-- Line 163: It is not completely clear how the 400 deposited seeds are translated to the 2.5D Voronoi structure. Can the authors describe in an exact and specific fashion what was done? That is - specifics of 3d tessellations, specific transformation, specific codes used.

-- Line 165: Do you mean that each cell had the same -yet scaled- polygon describing its 2D apical and basal surfaces? Please better explain.

-- Line 175: “comparable to a behavior law” – not clear.

-- Line 191: It is not clear what “closely as possible” means. Can the authors better describe how these parameters were set, and to what extent are the simulations sensitive to these?

-- Line 195: the title of table 2 is “Rheology model parameters…” but it basically contains only (small deformations) elasticity parameters. There are no viscosity or time-rate-dependent parameters.

-- Line 208: like in the introduction, it is not clearly described what is the FEM approach used here? What are the differences between older approaches and this suggested new approach?

-- Line 211: It is not clear what constraints are the source of the problem, as these constraints are repeatedly mentioned in the coming lines.

-- Line 212: “VM” - above Above it was termed AVM.

-- Line 218: At this point, I believe you mean the AVM model. It needs to be consistent throughout the text.

-- In Fig 3A please change the title to Active Vertex Model (AVM) - needed to maintain consistency.

--Line 222: You can also do it using a classic vertex model. You should emphasize that here, you also add the cell interior resistance, and by combining a continuous approach, you get a full stress tensor.

--Line 223: There are also other metrics to translate the local stress tensor to a local scalar, for example, the trace of the tensor (a way to quantify the average tension). What is the incentive to use von Mises stress? For example, in classical engineering, von Mises stress is convenient for formulating safety criteria.

--Line 230: the first sentence is redundant at this point in the MS.

--Line 232: Not clear. Do you mean an observed non-uniform curvature distribution along the organoid surface?

--Line 233: It is not clear what is “presents curvature”. Also, “no significant change in cell shape or thickness” is not clear. Do you mean no heterogeneity of cell shape or thickness?

--Line 244: “Solicitation” is a somewhat awkward word here. Do you mean “stimulation”? I would consider revisiting throughout the text and the relevant figures.

--Line 298: “temperature” can be confusing to many, and thus another reason to better explain the entire FEM approach here.

Minor comments:

-- In supp fig 2, in the right green box, change to “cells centers”.

-- Line 118: it should be the organoid shell (or epithelial monolayer) thickness.

-- Line 123 to 128: the sentence is too long.

-- In table 1: please define what is “thickness ratio”.

-- Line 144: Like was done in the figure caption- If you choose to specify the tool for 2D segmentation then you should probably also specify the tool for 3D reconstruction.

-- Line 170: do you mean” thickness ratio”?

--Line 238: What is FEA? Not previously defined

**Have the authors made all data and (if applicable) computational code underlying the findings in their manuscript fully available?**

Reviewer #1: **No: **The journal policy states that "code should be provided as part of the manuscript or its supporting information, or deposited to a public repository" but the authors state "to be made available upon reasonable request". I leave it up to the editors to decide if this is satisfactory.

Reviewer #3: **No: **See paragraph 3 in my review.

PLOS authors have the option to publish the peer review history of their article (what does this mean?). If published, this will include your full peer review and any attached files.

Reviewer #1: No

Reviewer #3: No
---

## [Editor Report · Decision Letter 2]

30 Oct 2024

PCOMPBIOL-D-24-00672R2Deciphering the interplay between biology and physics with a finite element method-implemented vertex organoid model: a tool for the mechanical analysis of cell behavior on a spherical organoid shell.PLOS Computational Biology Dear Dr. Ferrand, Thank you for submitting your manuscript to PLOS Computational Biology. After careful consideration, we feel that it has merit but does not fully meet PLOS Computational Biology's publication criteria as it currently stands. Therefore, we invite you to submit a revised version of the manuscript that addresses the points raised during the review process. Please submit your revised manuscript within 30 days Dec 30 2024 11:59PM. If you will need more time than this to complete your revisions, please reply to this message or contact the journal office at ploscompbiol@plos.org. Please include the following items when submitting your revised manuscript:*
A rebuttal letter that responds to each point raised by the editor and reviewer(s). You should upload this letter as a separate file labeled 'Response to Reviewers'. This file does not need to include responses to formatting updates and technical items listed in the 'Journal Requirements' section below.*
A marked-up copy of your manuscript that highlights changes made to the original version. You should upload this as a separate file labeled 'Revised Manuscript with Track Changes'.*
An unmarked version of your revised paper without tracked changes. You should upload this as a separate file labeled 'Manuscript'. If you would like to make changes to your financial disclosure, competing interests statement, or data availability statement, please make these updates within the submission form at the time of resubmission. Guidelines for resubmitting your figure files are available below the reviewer comments at the end of this letter. We look forward to receiving your revised manuscript. Kind regards, Nir GovAcademic EditorPLOS Computational Biology Pedro MendesSection EditorPLOS Computational Biology

Feilim Mac Gabhann

Editor-in-Chief

PLOS Computational Biology

Jason Papin

Editor-in-Chief

PLOS Computational Biology

 **Journal Requirements:** **Additional Editor Comments (if provided):**

I have read your answers to the reviewers, and I would like you to make the following minor revision before the paper is accepted for publication:

- Please incorporate all the answers to the reviewers in the revised manuscript, Such that you include the information that you give in the response letter.

- The new revised sentences are also in dire need of English editing.

**Reviewers' comments:**   **Figure resubmission:** While revising your submission, please upload your figure files to the Preflight Analysis and Conversion Engine (PACE) digital diagnostic tool, https://pacev2.apexcovantage.com/. PACE helps ensure that figures meet PLOS requirements. To use PACE, you must first register as a user. Registration is free. Then, login and navigate to the UPLOAD tab, where you will find detailed instructions on how to use the tool. If you encounter any issues or have any questions when using PACE, please email PLOS at figures@plos.org. Please note that Supporting Information files do not need this step. If there are other versions of figure files still present in your submission file inventory at resubmission, please replace them with the PACE-processed versions. **Reproducibility:** To enhance the reproducibility of your results, we recommend that authors of applicable studies deposit laboratory protocols in protocols.io, where a protocol can be assigned its own identifier (DOI) such that it can be cited independently in the future. Additionally, PLOS ONE offers an option to publish peer-reviewed clinical study protocols. Read more information on sharing protocols at https://plos.org/protocols?utm_medium=editorial-email&utm_source=authorletters&utm_campaign=protocols

---

## [Editor Report · Decision Letter 3]

27 Nov 2024

Dear Dr Ferrand,

We are pleased to inform you that your manuscript 'Deciphering the interplay between biology and physics with a finite element method-implemented vertex organoid model: a tool for the mechanical analysis of cell behavior on a spherical organoid shell.' has been provisionally accepted for publication in PLOS Computational Biology.

Best regards,

Nir Gov

Academic Editor

PLOS Computational Biology

Pedro Mendes

Section Editor

PLOS Computational Biology

Feilim Mac Gabhann

Editor-in-Chief

PLOS Computational Biology

Jason Papin

Editor-in-Chief

PLOS Computational Biology

I would ask the authors to address these minor issues for the final version:

1) Please give the English another editing round. For example: "Tests of arbitrarily chosen combinations of parameters also lead to most non-convergence of the calculation." this sentence needs editing.

2) In Table 4, the values of the Young Modulus for the T3D3T cells varies by many orders of magnitude between the basal and lateral cortex. How could this be justified ? does it simply mean that the lateral side is effectively very soft, but the exact number is not known ?

---

## [Editor Report · Acceptance letter]

18 Dec 2024

PCOMPBIOL-D-24-00672R3 

Deciphering the interplay between biology and physics with a finite element method-implemented vertex organoid model: a tool for the mechanical analysis of cell behavior on a spherical organoid shell.

Dear Dr Ferrand,

I am pleased to inform you that your manuscript has been formally accepted for publication in PLOS Computational Biology. Your manuscript is now with our production department and you will be notified of the publication date in due course.

With kind regards,

Anita Estes
